# Estimating the class prior and posterior from noisy positives and unlabeled data

**Shantanu Jain, Martha White, Predrag Radivojac**
Department of Computer Science
Indiana University, Bloomington, Indiana, USA
{shajain, martha, predrag}@indiana.edu

## Abstract

We develop a classification algorithm for estimating posterior distributions from positive-unlabeled data, that is robust to noise in the positive labels and effective for high-dimensional data. In recent years, several algorithms have been proposed to learn from positive-unlabeled data; however, many of these contributions remain theoretical, performing poorly on real high-dimensional data that is typically contaminated with noise. We build on this previous work to develop two practical classification algorithms that explicitly model the noise in the positive labels and utilize univariate transforms built on discriminative classifiers. We prove that these univariate transforms preserve the class prior, enabling estimation in the univariate space and avoiding kernel density estimation for high-dimensional data. The theoretical development and parametric and nonparametric algorithms proposed here constitute an important step towards wide-spread use of robust classification algorithms for positive-unlabeled data.

## 1 Introduction

Access to positive, negative and unlabeled examples is a standard assumption for most semi-supervised binary classification techniques. In many domains, however, a sample from one of the classes (say, negatives) may not be available, leading to the setting of learning from positive and unlabeled data (Denis et al., 2005). Positive-unlabeled learning often emerges in sciences and commerce where an observation of a positive example (say, that a protein catalyzes reactions or that a social network user likes a particular product) is usually reliable. Here, however, the absence of a positive observation cannot be interpreted as a negative example. In molecular biology, for example, an attempt to label a data point as positive (say, that a protein is an enzyme) may be unsuccessful for a variety of experimental and biological reasons, whereas in social networks an explicit dislike of a product may not be possible. Both scenarios lead to a situation where negative examples cannot be actively collected.

Fortunately, the absence of negatively labeled examples can be tackled by incorporating unlabeled examples as negatives, leading to the development of non-traditional classifiers. Here we follow the terminology by Elkan and Noto (2008) that a traditional classifier predicts whether an example is positive or negative, whereas a non-traditional classifier predicts whether the example is positive or unlabeled. Positive vs. unlabeled (non-traditional) training is reasonable because the class posterior — and also the optimum scoring function for composite losses (Reid and Williamson, 2010) — in the traditional setting is monotonically related to the posterior in the non-traditional setting. However, the true posterior can be fully recovered from the non-traditional posterior only if we know the class prior; i.e., the proportion of positives in unlabeled data. The knowledge of the class prior is also necessary for estimation of the performance criteria such as the error rate, balanced error rate or F-measure, and also for finding the right threshold for the non-traditional scoring function that leads to an optimal classifier with respect to some criteria (Menon et al., 2015).

Class prior estimation in a nonparametric setting has been actively researched in the past decade offering an extensive theory of identifiability (Ward et al., 2009; Blanchard et al., 2010; Scott et al., 2013; Jain et al., 2016) and a few practical solutions (Elkan and Noto, 2008; Ward et al., 2009; du Plessis and Sugiyama, 2014; Sanderson and Scott, 2014; Jain et al., 2016; Ramaswamy et al., 2016). Application of these algorithms to real data, however, is limited in that none of the proposed algorithms simultaneously deals with noise in the labels and practical estimation for high-dimensional data.

Much of the theory on learning class priors relies on the assumption that either the distribution of positives is known or that the positive sample is clean. In practice, however, labeled data sets contain class-label noise, where an unspecified amount of negative examples contaminates the positive sample. This is a realistic scenario in experimental sciences where technological advances enabled generation of high-throughput data at a cost of occasional errors. One example for this comes from the studies of proteins using analytical chemistry technology; i.e., mass spectrometry. For example, in the process of peptide identification (Steen and Mann, 2004), bioinformatics methods are usually set to report results with specified false discovery rate thresholds (e.g., 1%). Unfortunately, statistical assumptions in these experiments are sometimes violated thereby leading to substantial noise in reported results, as in the case of identifying protein post-translational modifications. Similar amounts of noise might appear in social networks such as Facebook, where some users select 'like', even when they do not actually like a particular post. Further, the only approach that does consider similar such noise (Scott et al., 2013) requires density estimation, which is known to be problematic for high-dimensional data.

In this work, we propose the first classification algorithm, with class prior estimation, designed particularly for high-dimensional data with noise in the labeling of positive data. We first formalize the problem of class prior estimation from noisy positive and unlabeled data. We extend the existing identifiability theory for class prior estimation from positive-unlabeled data to this noise setting. We then show that we can practically estimate class priors and the posterior distributions by first transforming the input space to a univariate space, where density estimation is reliable. We prove that these transformations preserve class priors and show that they correspond to training a non-traditional classifier. We derive a parametric algorithm and a nonparametric algorithm to learn the class priors. Finally, we carry out experiments on synthetic and real-life data and provide evidence that the new approaches are sound and effective.

## 2 Problem formulation

Consider a binary classification problem of mapping an input space $\mathcal{X}$ to an output space $\mathcal{Y} = \{0, 1\}$. Let $f$ be the true distribution of inputs. It can be represented as the following mixture

$$f(x) = \alpha f_1(x) + (1 - \alpha)f_0(x), \tag{1}$$

where $x \in \mathcal{X}$, $y \in \mathcal{Y}$, $f_y$ are distributions over $\mathcal{X}$ for the positive ($y = 1$) and negative ($y = 0$) class, respectively; and $\alpha \in [0, 1)$ is the class prior or the proportion of the positive examples in $f$. We will refer to a sample from $f$ as unlabeled data.

Let $g$ be the distribution of inputs for the labeled data. Because the labeled sample contains some mislabeled examples, the corresponding distribution is also a mixture of $f_1$ and a small proportion, say $1 - \beta$, of $f_0$. That is,

$$g(x) = \beta f_1(x) + (1 - \beta)f_0(x), \tag{2}$$

where $\beta \in (0, 1]$. Observe that both mixtures have the same components but different mixing proportions. The simplest scenario is that the mixing components $f_0$ and $f_1$ correspond to the class-conditional distributions $p(x|Y = 0)$ and $p(x|Y = 1)$, respectively. However, our approach also permits transformations of the input space $\mathcal{X}$, thus resulting in a more general setup.

The objective of this work is to study the estimation of the class prior $\alpha = p(Y = 1)$ and propose practical algorithms for estimating $\alpha$. The efficacy of this estimation is clearly tied to $\beta$, where as $\beta$ gets smaller, the noise in the positive labels becomes larger. We will discuss identifiability of $\alpha$ and $\beta$ and give a practical algorithm for estimating $\alpha$ (and $\beta$). We will then use these results to estimate the posterior distribution of the class variable, $p(y|x)$, despite the fact that the labeled set does not contain any negative examples.

## 3 Identifiability

The class prior is identifiable if there is a unique class prior for a given pair $(f, g)$. Much of the identifiability characterization in this section has already been considered as the case of asymmetric noise (Scott et al., 2013); see Section 7 on related work. We recreate these results here, with the aim to introduce required notation, to highlight several important results for later algorithm development and to include a few missing results needed for our approach. Though the proof techniques are themselves quite different and could be of interest, we include them in the appendix due to space.

There are typically two aspects to address with identifiability. First, one needs to determine if a problem is identifiable, and, second, if it is not, propose a canonical form that is identifiable. In this section we will see that class prior is not identifiable in general because $f_0$ can be a mixture containing $f_1$ and vice versa. To ensure identifiability, it is necessary to choose a canonical form that prefers a class prior that makes the two components as different as possible; this canonical form was introduced as the mutual irreducibility condition (Scott et al., 2013) and is related to the proper novelty distribution (Blanchard et al., 2010) and the max-canonical form (Jain et al., 2016).

We discuss identifiability in terms of measures. Let $\mu$, $\nu$, $\mu_0$ and $\mu_1$ be probability measures defined on some $\sigma$-algebra $\mathcal{A}$ on $\mathcal{X}$, corresponding to $f$, $g$, $f_0$ and $f_1$, respectively. It follows that

$$\mu = \alpha\mu_1 + (1 - \alpha)\mu_0 \tag{3}$$
$$\nu = \beta\mu_1 + (1 - \beta)\mu_0. \tag{4}$$

Consider a family of pairs of mixtures having the same components

$$\mathcal{F}(\Pi) = \{(\mu, \nu) : \mu = \alpha\mu_1 + (1 - \alpha)\mu_0, \nu = \beta\mu_1 + (1 - \beta)\mu_0, (\mu_0, \mu_1) \in \Pi, 0 \leq \alpha < \beta \leq 1\},$$

where $\Pi$ is some set of pairs of probability measures defined on $\mathcal{A}$. The family is parametrized by the quadruple $(\alpha, \beta, \mu_0, \mu_1)$. The condition $\beta > \alpha$ means that $\nu$ has a greater proportion of $\mu_1$ compared to $\mu$. This is consistent with our assumption that the labeled sample mainly contains positives. The most general choice for $\Pi$ is

$$\Pi^{\text{all}} = \mathcal{P}^{\text{all}} \times \mathcal{P}^{\text{all}} \setminus \{(\mu, \mu) : \mu \in \mathcal{P}^{\text{all}}\},$$

where $\mathcal{P}^{\text{all}}$ is the set of all probability measures defined on $\mathcal{A}$ and $\{(\mu, \mu) : \mu \in \mathcal{P}^{\text{all}}\}$ is the set of pairs with equal distributions. Removing equal pairs prevents $\mu$ and $\nu$ from being identical.

We now define the maximum proportion of a component $\lambda_1$ in a mixture $\lambda$, which is used in the results below and to specify the criterion that enables identifiability; more specifically,

$$a_\lambda^{\lambda_1} = \max\{\alpha \in [0, 1] : \lambda = \alpha\lambda_1 + (1 - \alpha)\lambda_0, \lambda_0 \in \mathcal{P}^{\text{all}}\}. \tag{5}$$

Of particular interest is the case when $a_\lambda^{\lambda_1} = 0$, which should be read as "$\lambda$ is not a mixture containing $\lambda_1$". We finally define the set all possible $(\alpha, \beta)$ that generate $\mu$ and $\nu$ when $(\mu_0, \mu_1)$ varies in $\Pi$:

$$A^+(\mu, \nu, \Pi) = \{(\alpha, \beta) : \mu = \alpha\mu_1 + (1 - \alpha)\mu_0, \nu = \beta\mu_1 + (1 - \beta)\mu_0, (\mu_0, \mu_1) \in \Pi, 0 \leq \alpha < \beta \leq 1\}.$$

If $A^+(\mu, \nu, \Pi)$ is a singleton set for all $(\mu, \nu) \in \mathcal{F}(\Pi)$, then $\mathcal{F}(\Pi)$ is identifiable in $(\alpha, \beta)$.

First, we show that the most general choice for $\Pi$, $\Pi^{\text{all}}$, leads to unidentifiability (Lemma 1). Fortunately, however, by choosing a restricted set

$$\Pi^{\text{res}} = \{(\mu_0, \mu_1) \in \Pi^{\text{all}} : a_{\mu_0}^{\mu_1} = 0, a_{\mu_1}^{\mu_0} = 0\}$$

as $\Pi$, we do obtain identifiability (Theorem 1). In words, $\Pi^{\text{res}}$ contains pairs of distributions, where each distribution in a pair cannot be expressed as a mixture containing the other. The proofs of the results below are in the Appendix.

**Lemma 1 (Unidentifiability)** *Given a pair of mixtures $(\mu, \nu) \in \mathcal{F}(\Pi^{all})$, let parameters $(\alpha, \beta, \mu_0, \mu_1)$ generate $(\mu, \nu)$ and $\alpha^+ = a_\mu^\nu, \beta^+ = a_\nu^\mu$. It follows that*

    *1. There is a one-to-one relation between $(\mu_0, \mu_1)$ and $(\alpha, \beta)$ and*

$$\mu_0 = \frac{\beta\mu - \alpha\nu}{\beta - \alpha}, \qquad \mu_1 = \frac{(1 - \alpha)\nu - (1 - \beta)\mu}{\beta - \alpha}. \tag{6}$$

2. *Both expressions on the right-hand side of Equation 6 are well defined probability measures if and only if $\alpha/\beta \leq \alpha^+$ and $(1-\beta)/(1-\alpha) \leq \beta^+$.*

3. $A^+(\mu, \nu, \Pi^{all}) = \{(\alpha, \beta) : \alpha/\beta \leq \alpha^+, (1-\beta)/(1-\alpha) \leq \beta^+\}.$

4. *$\mathcal{F}(\Pi^{all})$ is unidentifiable in $(\alpha, \beta)$; i.e., $(\alpha, \beta)$ is not uniquely determined from $(\mu, \nu)$.*

5. *$\mathcal{F}(\Pi^{all})$ is unidentifiable in $\alpha$ and $\beta$, individually; i.e., neither $\alpha$ nor $\beta$ is uniquely determined from $(\mu, \nu)$.*

Observe that the definition of $a_\lambda^{\lambda_1}$ and $\mu \neq \nu$ imply $\alpha^+ < 1$ and, consequently, any $(\alpha, \beta) \in A^+(\mu, \nu, \Pi^{all})$ satisfies $\alpha < \beta$, as expected.

**Theorem 1 (Identifiablity)** *Given $(\mu, \nu) \in \mathcal{F}(\Pi^{all})$, let $\alpha^+ = a_\mu^\nu$ and $\beta^+ = a_\nu^\mu$. Let $\mu_0^* = (\mu - \alpha^+ \nu)/(1 - \alpha^+)$, $\mu_1^* = (\nu - \beta^+ \mu)/(1 - \beta^+)$ and*

$$\alpha^* = \alpha^+(1 - \beta^+)/(1 - \alpha^+ \beta^+), \qquad \beta^* = (1 - \beta^+)/(1 - \alpha^+ \beta^+). \tag{7}$$

*It follows that*

1. *$(\alpha^*, \beta^*, \mu_0^*, \mu_1^*)$ generate $(\mu, \nu)$*

2. *$(\mu_0^*, \mu_1^*) \in \Pi^{res}$ and, consequently, $\alpha^* = a_\mu^{\mu_1^*}$, $\beta^* = a_\nu^{\mu_1^*}$.*

3. *$\mathcal{F}(\Pi^{res})$ contains all pairs of mixtures in $\mathcal{F}(\Pi^{all})$.*

4. *$A^+(\mu, \nu, \Pi^{res}) = \{(\alpha^*, \beta^*)\}$.*

5. *$\mathcal{F}(\Pi^{res})$ is identifiable in $(\alpha, \beta)$; i.e., $(\alpha, \beta)$ is uniquely determined from $(\mu, \nu)$.*

We refer to the expressions of $\mu$ and $\nu$ as mixtures of components $\mu_0$ and $\mu_1$ as a max-canonical form when $(\mu_0, \mu_1)$ is picked from $\Pi^{res}$. This form enforces that $\mu_1$ is not a mixture containing $\mu_0$ and vice versa, which leads to $\mu_0$ and $\mu_1$ having maximum separation, while still generating $\mu$ and $\nu$. Each pair of distributions in $\mathcal{F}(\Pi^{res})$ is represented in this form. Identifiability of $\mathcal{F}(\Pi^{res})$ in $(\alpha, \beta)$ occurs precisely when $A^+(\mu, \nu, \Pi^{res}) = \{(\alpha^*, \beta^*)\}$, i.e., $(\alpha^*, \beta^*)$ is the only pair of mixing proportions that can appear in a max-canonical form of $\mu$ and $\nu$. Moreover, Statement 1 in Theorem 1 and Statement 1 in Lemma 1 imply that the max-canonical form is unique and completely specified by $(\alpha^*, \beta^*, \mu_0^*, \mu_1^*)$, with $\alpha^* < \beta^*$ following from Equation 7. Thus, using $\mathcal{F}(\Pi^{res})$ to model the unlabeled and labeled data distributions makes estimation of not only $\alpha$, the class prior, but also $\beta, \mu_0, \mu_1$ a well-posed problem. Moreover, due to Statement 3 in Theorem 1, there is no loss in the modeling capability by using $\mathcal{F}(\Pi^{res})$ instead of $\mathcal{F}(\Pi^{all})$. Overall, identifiability, absence of loss of modeling capability and maximum separation between $\mu_0$ and $\mu_1$ combine to justify estimating $\alpha^*$ as the class prior.

# 4 Univariate Transformation

The theory and algorithms for class prior estimation are agnostic to the dimensionality of the data; in practice, however, this dimensionality can have important consequences. Parametric Gaussian mixture models trained via expectation-maximization (EM) are known to strongly suffer from co-linearity in high-dimensional data. Nonparametric (kernel) density estimation is also known to have curse-of-dimensionality issues, both in theory (Liu et al., 2007) and in practice (Scott, 2008).

We address the curse of dimensionality by transforming the data to a single dimension. The transformation $\tau : \mathcal{X} \to \mathbb{R}$, surprisingly, is simply an output of a non-traditional classifier trained to separate labeled sample, $L$, from unlabeled sample, $U$. The transform is similar to that in (Jain et al., 2016), except that it is not required to be calibrated like a posterior distribution; as shown below, a good ranking function is sufficient. First, however, we introduce notation and formalize the data generation steps (Figure 1).

Let $X$ be a random variable taking values in $\mathcal{X}$, capturing the true distribution of inputs, $\mu$, and $Y$ be an unobserved random variable taking values in $\mathcal{Y}$, giving the true class of the inputs. It follows that $X|Y = 0$ and $X|Y = 1$ are distributed according to $\mu_0$ and $\mu_1$, respectively. Let $S$ be a selection random variable, whose value in $\mathcal{S} = \{0, 1, 2\}$ determines the sample to which an input $x$ is added (Figure 1). When $S = 1$, $x$ is added to the noisy labeled sample; when $S = 0$, $x$ is added to the unlabeled sample; and when $S = 2$, $x$ is not added to either of the samples. It follows that

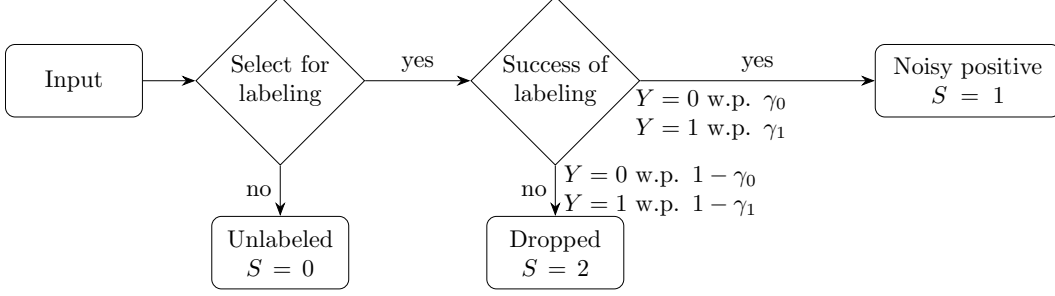

Figure 1: The labeling procedure, with $S$ taking values from $\mathcal{S} = \{0, 1, 2\}$. In the first step, the sample is randomly selected to attempt labeling, with some probability independent of $X$ or $Y$. If it is not selected, it is added to the "Unlabeled" set. If it is selected, then labeling is attempted. If the true label is $Y = 1$, then with probability $\gamma_1 \in (0, 1)$, the labeling will succeed and it is added to "Noisy positives". Otherwise, it is added to the "Dropped" set. If the true label is $Y = 0$, then the attempted labeling is much more likely to fail, but because of noise, could succeed. The attempted label of $Y = 0$ succeeds with probability $\gamma_0$, and is added to "Noisy positives", even though it is actually a negative instance. $\gamma_0 = 0$ leads to the no noise case and the noise increases as $\gamma_0$ increases. $\beta = {\gamma_1 \alpha}/{(\gamma_1 \alpha + \gamma_0 (1 - \alpha))}$, gives the proportion of positives in the "Noisy positives".

$X^u = X|S = 0$ and $X^l = X|S = 1$ are distributed according to $\mu$ and $\nu$, respectively. We make the following assumptions which are consistent with the statements above:

$$p(y|S = 0) = p(y), \qquad (8)$$
$$p(y = 1|S = 1) = \beta, \qquad (9)$$
$$p(x|s, y) = p(x|y). \qquad (10)$$

Assumptions 8 and 9 states that the proportion of positives in the unlabeled sample and the labeled sample matches the true proportion in $\mu$ and $\nu$, respectively. Assumption 10 states that the distribution of the positive inputs (and the negative inputs) in both the unlabeled and the labeled samples is equal and unbiased. Lemma 2 gives the implications of these assumptions. Statement 3 in Lemma 2 is particularly interesting and perhaps counter-intuitive as it states that with non-zero probability some inputs need to be dropped.

**Lemma 2** *Let $X$, $Y$ and $S$ be random variables taking values in $\mathcal{X}$, $\mathcal{Y}$ and $\mathcal{S}$, respectively, and $X^u = X|S = 0$ and $X^l = X|S = 1$. For measures $\mu, \nu, \mu_0, \mu_1$, satisfying Equations 3 and 4 and $\mu_1 \neq \mu_0$, let $\mu, \mu_0, \mu_1$ give the distribution of $X$, $X|Y = 0$ and $X|Y = 1$, respectively. If $X, Y$ and $S$ satisfy assumptions 8, 9 and 10, then*

1. *$X$ is independent of $S = 0$; i.e., $p(x|S = 0) = p(x)$*
2. *$X^u$ and $X^l$ are distributed according to $\mu$ and $\nu$, respectively.*
3. *$p(S = 2) \neq 0$.*

The proof is in the Appendix. Next, we highlight the conditions under which the score function $\tau$ preserves $\alpha^*$. Observing that $S$ serves as the pseudo class label for labeled vs. unlabeled classification as well, we first give an expression for the posterior:

$$\tau_p(x) = p(S = 1|x, S \in \{0, 1\}), \ \forall x \in \mathcal{X}. \qquad (11)$$

**Theorem 2 ($\alpha^*$-preserving transform)** *Let random variables $X, Y, S, X^u, X^l$ and measures $\mu, \nu, \mu_0, \mu_1$ be as defined in Lemma 2. Let $\tau_p$ be the posterior as defined in Equation 11 and $\tau = H \circ \tau_p$, where $H$ is a 1-to-1 function on $[0, 1]$ and $\circ$ is the composition operator. Assume*

1. *$(\mu_0, \mu_1) \in \Pi^{res}$,*
2. *$X^u$ and $X^l$ are continuous with densities $f$ and $g$, respectively,*
3. *$\mu_\tau, \nu_\tau, \mu_{\tau 1}$ are the measures corresponding to $\tau(X^u), \tau(X^l), \tau(X_1)$, respectively,*
4. *$(\alpha^+, \beta^+, \alpha^*, \beta^*) = (a_\mu^\nu, a_\nu^\mu, a_\mu^{\mu_1}, a_\nu^{\mu_1})$ and $(\alpha_\tau^+, \beta_\tau^+, \alpha_\tau^*, \beta_\tau^*) = (a_{\mu_\tau}^{\nu_\tau}, a_{\nu_\tau}^{\mu_\tau}, a_{\mu_\tau}^{\mu_{\tau 1}}, a_{\nu_\tau}^{\mu_{\tau 1}})$.*

*Then*

$$(\alpha_\tau^+, \beta_\tau^+, \alpha_\tau^*, \beta_\tau^*) = (\alpha^+, \beta^+, \alpha^*, \beta^*)$$

*and so $\tau$ is an $\alpha^*$-preserving transformation.*

*Moreover, $\tau_p$ can also be used to compute the true posterior probability:*

$$p(Y = 1|x) = \frac{\alpha^*(1 - \alpha^*)}{\beta^* - \alpha^*} \left( \frac{p(S = 0)}{p(S = 1)} \frac{\tau_p(x)}{1 - \tau_p(x)} - \frac{1 - \beta^*}{1 - \alpha^*} \right). \qquad (12)$$

The proof is in the Appendix. Theorem 2 shows that the $\alpha^*$ is the same for the original data and the transformed data, if the transformation function $\tau$ can be expressed as a composition of $\tau_p$ and a one-to-one function, $H$, defined on $[0, 1]$. Trivially, $\tau_p$ itself is one such function. We emphasize, however, that $\alpha^*$-preservation is not limited by the efficacy of the calibration algorithm; uncalibrated scoring that ranks inputs as $\tau_p(x)$ also preserves $\alpha^*$. Theorem 2 further demonstrates how the true posterior, $p(Y = 1|x)$, can be recovered from $\tau_p$ by plugging in estimates of $\tau_p$, $p(S=0)/p(S=1)$, $\alpha^*$ and $\beta^*$ in Equation 12. The posterior probability $\tau_p$ can be estimated directly by using a probabilistic classifier or by calibrating a classifier's score (Platt, 1999; Niculescu-Mizil and Caruana, 2005); $|U|/|L|$ serves as an estimate of $p(S=0)/p(S=1)$; section 5 gives parametric and nonparametric approaches for estimation of $\alpha^*$ and $\beta^*$.

## 5  Algorithms

In this section, we derive a parametric and a nonparametric algorithm to estimate $\alpha^*$ and $\beta^*$ from the unlabeled sample, $U = \{X_i^u\}$, and the noisy positive sample, $L = \{X_i^l\}$. In theory, both approaches can handle multivariate samples; in practice, however, to circumvent the curse of dimensionality, we exploit the theory of $\alpha^*$-preserving univariate transforms to transform the samples.

**Parametric approach.** The parametric approach is derived by modeling each sample as a two component Gaussian mixture, sharing the same components but having different mixing proportions:

$$X_i^u \sim \alpha \mathcal{N}(u_1, \Sigma_1) + (1 - \alpha)\mathcal{N}(u_0, \Sigma_0)$$
$$X_i^l \sim \beta \mathcal{N}(u_1, \Sigma_1) + (1 - \beta)\mathcal{N}(u_0, \Sigma_0)$$

where $u_1, u_0 \in \mathbb{R}^d$ and $\Sigma_1, \Sigma_0 \in \mathbb{S}_{++}^d$, the set of all $d \times d$ positive definite matrices. The algorithm is an extension to the EM approach for Gaussian mixture models (GMMs) where, instead of estimating the parameters of a single mixture, the parameters of both mixtures $(\alpha, \beta, u_0, u_1, \Sigma_0, \Sigma_1)$ are estimated simultaneously by maximizing the combined likelihood over both $U$ and $L$. This approach, which we refer to as a multi-sample GMM (MSGMM), exploits the constraint that the two mixtures share the same components. The update rules and their derivation are given in the Appendix.

**Nonparametric approach.** Our nonparametric strategy directly exploits the results of Lemma 1 and Theorem 1, which give a direct connection between $(\alpha^+ = a_\mu^\nu, \beta^+ = a_\nu^\mu)$ and $(\alpha^*, \beta^*)$. Therefore, for a two-component mixture sample, $M$, and a sample from one of the components, $C$, it only requires an algorithm to estimate the maximum proportion of $C$ in $M$. For this purpose, we use the AlphaMax algorithm (Jain et al., 2016), briefly summarized in the Appendix. Specifically, our two-step approach for estimating $\alpha^*$ and $\beta^*$ is as follows: $(i)$ Estimate $\alpha^+$ and $\beta^+$ as outputs of AlphaMax$(U, L)$ and AlphaMax$(L, U)$, respectively; $(ii)$ Estimate $(\alpha^*, \beta^*)$ from the estimates of $(\alpha^+, \beta^+)$ by applying Equation 7. We refer to our nonparametric algorithm as AlphaMax-N.

## 6  Empirical investigation

In this section we systematically evaluate the new algorithms in a controlled, synthetic setting as well as on a variety of data sets from the UCI Machine Learning Repository (Lichman, 2013).

**Experiments on synthetic data:** We start by evaluating all algorithms in a univariate setting where both mixing proportions, $\alpha$ and $\beta$, are known. We generate unit-variance Gaussian and unit-scale Laplace-distributed i.i.d. samples and explore the impact of mixing proportions, the size of the component sample, and the separation and overlap between the mixing components on the accuracy of estimation. The class prior $\alpha$ was varied from $\{0.05, 0.25, 0.50\}$ and the noise component $\beta$ from $\{1.00, 0.95, 0.75\}$. The size of the labeled sample $L$ was varied from $\{100, 1000\}$, whereas the size of the unlabeled sample $U$ was fixed at 10000.

**Experiments on real-life data:** We considered twelve real-life data sets from the UCI Machine Learning Repository. To adjust these data to our problems, categorical features were transformed into numerical using sparse binary representation, the regression data sets were transformed into classification based on mean of the target variable, and the multi-class classification problems were converted into binary problems by combining classes. In each data set, a subset of positive and negative examples was randomly selected to provide a labeled sample while the remaining data (without class labels) were used as unlabeled data. The size of the labeled sample was kept at 1000 (or 100 for small data sets) and the maximum size of unlabeled data was set 10000.

**Algorithms:** We compare the AlphaMax-N and MSGMM algorithms to the Elkan-Noto algorithm (Elkan and Noto, 2008) as well as the noiseless version of AlphaMax (Jain et al., 2016). There are several versions of the Elkan-Noto estimator and each can use any underlying classifier. We used the $e_1$ alternative estimator combined with the ensembles of 100 two-layer feed-forward neural networks, each with five hidden units. The out-of-bag scores of the same classifier were used as a class-prior preserving transformation that created an input to the AlphaMax algorithms. It is important to mention that neither Elkan-Noto nor AlphaMax algorithm was developed to handle noisy labeled data. In addition, the theory behind the Elkan-Noto estimator restricts its use to class-conditional distributions with non-overlapping supports. The algorithm by du Plessis and Sugiyama (2014) minimizes the same objective as the $e_1$ Elkan-Noto estimator and, thus, was not implemented.

**Evaluation:** All experiments were repeated 50 times to be able to draw conclusions with statistical significance. In real-life data, the labeled sample was created randomly by choosing an appropriate number of positive and negative examples to satisfy the condition for $\beta$ and the size of the labeled sample, while the remaining data was used as the unlabeled sample. Therefore, the class prior in the unlabeled data varies with the selection of the noise parameter $\beta$. The mean absolute difference between the true and estimated class priors was used as a performance measure. The best performing algorithm on each data set was determined by multiple hypothesis testing using the P-value of $0.05$ and Bonferroni correction.

**Results:** The comprehensive results for synthetic data drawn from univariate Gaussian and Laplace distributions are shown in Appendix (Table 2). In these experiments no transformation was applied prior to running any of the algorithms. As expected, the results show excellent performance of the MSGMM model on the Gaussian data. These results significantly degrade on Laplace-distributed data, suggesting sensitivity to the underlying assumptions. On the other hand, AlphaMax-N was accurate over all data sets and also robust to noise. These results suggest that new parametric and nonparametric algorithms perform well in these controlled settings.

Table 1 shows the results on twelve real data sets. Here, AlphaMax and AlphaMax-N algorithms demonstrate significant robustness to noise, although the parametric version MSGMM was competitive in some cases. On the other hand, the Elkan-Noto algorithm expectedly degrades with noise. Finally, we investigated the practical usefulness of the $\alpha^*$-preserving transform. Table 3 (Appendix) shows the results of AlphaMax-N and MSGMM on the real data sets, with and without using the transform. Because of computational and numerical issues, we reduced the dimensionality by using principal component analysis (the original data caused matrix singularity issues for MSGMM and density estimation issues for AlphaMax-N). MSGMM deteriorates significantly without the transform, whereas AlphaMax-N preserves some signal for the class prior. AlphaMax-N with the transform, however, shows superior performance on most data sets.

# 7 Related work

Class prior estimation in a semi-supervised setting, including positive-unlabeled learning, has been extensively discussed previously; see Saerens et al. (2002); Cortes et al. (2008); Elkan and Noto (2008); Blanchard et al. (2010); Scott et al. (2013); Jain et al. (2016) and references therein. Recently, a general setting for label noise has also been introduced, called the mutual contamination model. The aim under this model is to estimate multiple unknown base distributions, using multiple random samples that are composed of different convex combinations of those base distributions (Katz-Samuels and Scott, 2016). The setting of asymmetric label noise is a subset of this more general setting, treated under general conditions by Scott et al. (2013), and previously investigated under a more restrictive setting as co-training (Blum and Mitchell, 1998). A natural approach is to use robust estimation to learn in the presence of class noise; this strategy, however, has been shown to be ineffective, both theoretically (Long and Servedio, 2010; Manwani and Sastry, 2013) and empirically (Hawkins and McLachlan, 1997; Bashir and Carter, 2005), indicating the need to explicitly model the noise. Generative mixture model approaches have also been developed, which explicitly model the noise (Lawrence and Scholkopf, 2001; Bouveyron and Girard, 2009); these algorithms, however, assume labeled data for each class. As the most related work, though Scott et al. (2013) did not explicitly treat the positive-unlabeled learning with noisy positives, their formulation can incorporate this setting by using $\pi_0 = \alpha$ and $\beta = 1 - \pi_1$. The theoretical and algorithmic treatment, however, is very different. Their focus is on identifiability and analyzing convergence rates and statistical properties, assuming access to some $\kappa^*$ function which can obtain proportions

Table 1: Mean absolute difference between estimated and true mixing proportion over twelve data sets from the UCI Machine Learning Repository. Statistical significance was evaluated by comparing the Elkan-Noto algorithm, AlphaMax, AlphaMax-N, and the multi-sample GMM after applying a multivariate-to-univariate transform (MSGMM-T). The bold font type indicates the winner and the asterisk indicates statistical significance. For each data set, shown are the true mixing proportion ($\alpha$), true proportion of the positives in the labeled sample ($\beta$), sample dimensionality ($d$), the number of positive examples ($n_1$), the total number of examples ($n$), and the area under the ROC curve (AUC) for a model trained between labeled and unlabeled data.

| Data | $\alpha$ | $\beta$ | AUC | $d$ | $n_1$ | $n$ | Elkan-Noto | AlphaMax | AlphaMax-N | MSGMM-T |
|---|---|---|---|---|---|---|---|---|---|---|
| Bank | 0.095 | 1.00 | 0.842 | 13 | 5188 | 45000 | 0.241 | 0.070 | **0.037**\* | 0.163 |
| | 0.096 | 0.95 | 0.819 | 13 | 5188 | 45000 | 0.284 | 0.079 | **0.036**\* | 0.155 |
| | 0.101 | 0.75 | 0.744 | 13 | 5188 | 45000 | 0.443 | 0.124 | **0.040**\* | 0.127 |
| Concrete | 0.419 | 1.00 | 0.685 | 8 | 490 | 1030 | 0.329 | 0.141 | 0.181 | **0.077**\* |
| | 0.425 | 0.95 | 0.662 | 8 | 490 | 1030 | 0.363 | 0.174 | 0.231 | **0.095**\* |
| | 0.446 | 0.75 | 0.567 | 8 | 490 | 1030 | 0.531 | **0.212** | 0.272 | 0.233 |
| Gas | 0.342 | 1.00 | 0.825 | 127 | 2565 | 5574 | 0.017 | 0.011 | 0.017 | **0.008**\* |
| | 0.353 | 0.95 | 0.795 | 127 | 2565 | 5574 | 0.078 | 0.016 | **0.006** | 0.006 |
| | 0.397 | 0.75 | 0.672 | 127 | 2565 | 5574 | 0.396 | 0.137 | 0.009 | **0.006**\* |
| Housing | 0.268 | 1.00 | 0.810 | 13 | 209 | 506 | 0.159 | **0.087** | 0.094 | 0.209 |
| | 0.281 | 0.95 | 0.777 | 13 | 209 | 506 | 0.226 | **0.094** | 0.110 | 0.204 |
| | 0.330 | 0.75 | 0.651 | 13 | 209 | 506 | 0.501 | **0.125** | 0.134 | 0.172 |
| Landsat | 0.093 | 1.00 | 0.933 | 36 | 1508 | 6435 | 0.074 | 0.009 | **0.007**\* | 0.157 |
| | 0.103 | 0.95 | 0.904 | 36 | 1508 | 6435 | 0.110 | 0.015 | **0.008**\* | 0.152 |
| | 0.139 | 0.75 | 0.788 | 36 | 1508 | 6435 | 0.302 | 0.063 | **0.012**\* | 0.143 |
| Mushroom | 0.409 | 1.00 | 0.792 | 126 | 3916 | 8124 | 0.029 | **0.015**\* | 0.022 | 0.037 |
| | 0.416 | 0.95 | 0.766 | 126 | 3916 | 8124 | 0.087 | 0.015 | **0.008**\* | 0.037 |
| | 0.444 | 0.75 | 0.648 | 126 | 3916 | 8124 | 0.370 | 0.140 | **0.020** | 0.024 |
| Pageblock | 0.086 | 1.00 | 0.885 | 10 | 560 | 5473 | 0.116 | **0.026**\* | 0.044 | 0.129 |
| | 0.087 | 0.95 | 0.858 | 10 | 560 | 5473 | 0.137 | **0.031**\* | 0.052 | 0.125 |
| | 0.090 | 0.75 | 0.768 | 10 | 560 | 5473 | 0.256 | **0.041**\* | 0.064 | 0.111 |
| Pendigit | 0.243 | 1.00 | 0.875 | 16 | 3430 | 10992 | 0.030 | **0.006**\* | 0.009 | 0.081 |
| | 0.248 | 0.95 | 0.847 | 16 | 3430 | 10992 | 0.071 | 0.011 | **0.005**\* | 0.074 |
| | 0.268 | 0.75 | 0.738 | 16 | 3430 | 10992 | 0.281 | 0.093 | **0.007**\* | 0.062 |
| Pima | 0.251 | 1.00 | 0.735 | 8 | 268 | 768 | 0.351 | 0.120 | **0.111** | 0.171 |
| | 0.259 | 0.95 | 0.710 | 8 | 268 | 768 | 0.408 | 0.118 | **0.110** | 0.168 |
| | 0.289 | 0.75 | 0.623 | 8 | 268 | 768 | 0.586 | **0.144** | 0.156 | 0.175 |
| Shuttle | 0.139 | 1.00 | 0.929 | 9 | 8903 | 58000 | **0.024**\* | 0.027 | 0.029 | 0.157 |
| | 0.140 | 0.95 | 0.903 | 9 | 8903 | 58000 | 0.052 | **0.004**\* | 0.007 | 0.157 |
| | 0.143 | 0.75 | 0.802 | 9 | 8903 | 58000 | 0.199 | 0.047 | **0.004**\* | 0.148 |
| Spambase | 0.226 | 1.00 | 0.842 | 57 | 1813 | 4601 | 0.184 | 0.046 | **0.041** | 0.059 |
| | 0.240 | 0.95 | 0.812 | 57 | 1813 | 4601 | 0.246 | 0.059 | **0.042**\* | 0.063 |
| | 0.295 | 0.75 | 0.695 | 57 | 1813 | 4601 | 0.515 | 0.155 | **0.044**\* | 0.059 |
| Wine | 0.566 | 1.00 | 0.626 | 11 | 4113 | 6497 | 0.290 | 0.083 | **0.060** | 0.070 |
| | 0.575 | 0.95 | 0.610 | 11 | 4113 | 6497 | 0.322 | 0.113 | **0.063** | 0.076 |
| | 0.612 | 0.75 | 0.531 | 11 | 4113 | 6497 | 0.420 | 0.322 | 0.353 | **0.293** |

between samples. They do not explicitly address issues with high-dimensional data nor focus on algorithms to obtain $\kappa^*$. In contrast, we focus primarily on the univariate transformation to handle high-dimensional data and practical algorithms for estimating $\alpha^*$. Supervised learning used for class prior-preserving transformation provides a rich set of techniques to address high-dimensional data.

# 8 Conclusion

In this paper, we developed a practical algorithm for classification of positive-unlabeled data with noise in the labeled data set. In particular, we focused on a strategy for high-dimensional data, providing a univariate transform that reduces the dimension of the data, preserves the class prior so that estimation in this reduced space remains valid and is then further useful for classification. This approach provides a simple algorithm that simultaneously improves estimation of the class prior and provides a resulting classifier. We derived a parametric and a nonparametric version of the algorithm and then evaluated its performance on a wide variety of learning scenarios and data sets. To the best of our knowledge, this algorithm represents one of the first practical and easy-to-use approaches to learning with high-dimensional positive-unlabeled data with noise in the labels.

**Acknowledgements**

We thank Prof. Michael W. Trosset for helpful comments. Grant support: NSF DBI-1458477, NIH R01MH105524, NIH R01GM103725, and the Indiana University Precision Health Initiative.

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
