[Supplementary Material · jain_appendix.pdf]

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

# Appendix

## A    Identifiability proofs

We will need the following Lemma for the proofs.

**Lemma A.1** *Let $\mu$, $\mu_1$ and $\mu_0$ be three measures defined on $\mathcal{A}$ such that $\mu = \alpha\mu_1 + (1-\alpha)\mu_0$ for some $\alpha \in [0,1]$. Define*

$$R(\mu, \mu_1) = \{\mu(A)/\mu_1(A) : A \in \mathcal{A}, \mu_1(A) > 0\}.$$

*It follows that*

    *1. $a_\mu^{\mu_1} = \inf R(\mu, \mu_1)$.*

    *2. If $a_\mu^{\mu_1}_0 = 0$, then $\alpha = a_\mu^{\mu_1}$.*

**Proof:**  The proof follows from Lemma 4 and Theorem 3 of Jain et al. (2016). ∎

As a consequence of Statement 1 in Lemma A.1, we use an alternate characterization of $a_\lambda^{\lambda_1}$

$$a_\lambda^{\lambda_1} = \inf R(\lambda, \lambda_1) \tag{13}$$

where $R(\lambda, \lambda_1) = \{\lambda(A)/\lambda_1(A) : A \in \mathcal{A}, \lambda_1(A) > 0\}$.

**Lemma 1 (Unidentifiability)** *Given a pair of mixtures $(\mu, \nu) \in \mathcal{F}(\Pi^{all})$, let parameters $(\alpha, \beta, \mu_0, \mu_1)$ generate $(\mu, \nu)$ and $\alpha^+ = a_\mu^\nu, \beta^+ = a_\nu^\mu$. It follows that*

    *1. There is a one-to-one relation between $(\mu_0, \mu_1)$ and $(\alpha, \beta)$ and*

$$\mu_0 = \frac{\beta\mu - \alpha\nu}{\beta - \alpha}, \qquad \mu_1 = \frac{(1-\alpha)\nu - (1-\beta)\mu}{\beta - \alpha}. \tag{6}$$

    *2. Both expressions on the right-hand side of Equation 6 are well defined probability measures if and only if $\alpha/\beta \le \alpha^+$ and $(1-\beta)/(1-\alpha) \le \beta^+$.*

    *3. $A^+(\mu, \nu, \Pi^{all}) = \{(\alpha, \beta) : \alpha/\beta \le \alpha^+, (1-\beta)/(1-\alpha) \le \beta^+\}$.*

    *4. $\mathcal{F}(\Pi^{all})$ is unidentifiable in $(\alpha, \beta)$; i.e., $(\alpha, \beta)$ is not uniquely determined from $(\mu, \nu)$.*

    *5. $\mathcal{F}(\Pi^{all})$ is unidentifiable in $\alpha$ and $\beta$, individually; i.e., neither $\alpha$ nor $\beta$ is uniquely determined from $(\mu, \nu)$.*

**Proof:**
**Statement 1:**    Because $(\alpha, \beta, \mu_0, \mu_1)$ generate $\mu$ and $\nu$, $\beta > \alpha$ and Equations 3 and 4 hold. $\mu_0$ can be expressed in terms of $\mu, \nu, \alpha, \beta$ by eliminating $\mu_1$ ($\beta\times$ (Equation 3) $-\alpha\times$ (Equation 4)). Similarly $\mu_1$ can be expressed in terms of $\mu, \nu, \alpha, \beta$ and Equation 6 follows. Thus, given $\mu$ and $\nu$, $\mu_0, \mu_1$ can be uniquely determined from $\alpha, \beta$. Equation 3 implies that for all $A \in \mathcal{A}$ such that $\mu_1(A) - \mu_0(A) > 0$, $\alpha = (\mu(A)-\mu_1(A))/(\mu_1(A)-\mu_0(A))$. Existence of $A$ is guaranteed because $\mu_1 \ne \mu_0$ from the definition of $\Pi^{all}$. Thus, given $\mu$, $\alpha$ is uniquely determined from $\mu_0, \mu_1$. Similarly, Equation 4 implies that, given $\nu$, $\beta$ is uniquely determined from $\mu_0, \mu_1$.

**Statement 2:**    It is easy to observe that $(\beta\mu-\alpha\nu)/(\beta-\alpha)$ is a valid probability measure, if and only if $\beta\mu(A) - \alpha\nu(A) \ge 0$ for all $A \in \mathcal{A}$. Because the inequality is trivially true when $\nu(A) = 0$, the necessary and sufficient condition can be reduced to $\beta\mu(A) - \alpha\nu(A) \ge 0$ for all $A \in \mathcal{A}$ with $\nu(A) > 0$. This can be rewritten as $\alpha/\beta \le \mu(A)/\nu(A)$ for all $A \in \mathcal{A}$ such that $\nu(A) > 0$. Equivalently, $\alpha/\beta$ is a lower bound to $R(\mu, \nu)$; in other words, $\alpha/\beta \le \alpha^+$ due to the alternate characterization given by equation 13. Similarly, $((1-\alpha)\nu-(1-\beta)\mu)/(\beta-\alpha)$ is a valid probability measure provided $(1-\beta)/(1-\alpha) \le \beta^+$.

**Statement 3:**    Because system of equations in 6 is essentially equivalent to equations 3 and 4, $(\alpha, \beta) \in A^+(\mu, \nu, \Pi^{all})$ if and only if, $\mu_0$ and $\mu_1$ as defined in equations 6 come from $\Pi^{all}$. When the conditions of statement 2 are not satisfied, $\mu_0, \mu_1$ are not well-defined probability measures and

consequently $(\mu_0, \mu_1) \notin \Pi^{\text{all}}$. On the other hand, when they are satisfied, $\mu_0$, $\mu_1$ are well-defined probability measures. Moreover, $\mu \neq \nu$ implies $\mu_0 \neq \mu_1$ and consequently $(\mu_0, \mu_1) \in \Pi^{\text{all}}$.

**Statement 5:** We construct mixtures $\ddot{\mu}$ and $\ddot{\nu}$ such that there are multiple values for $\alpha$ that generate the mixtures. For $0 < \ddot{\alpha} < \ddot{\beta} < 1$ and probability measures $\ddot{\mu}_0, \ddot{\mu}_1$ on $\mathcal{A}$, let $(\ddot{\alpha}, \ddot{\beta}, \ddot{\mu}_0, \ddot{\mu}_1)$ generate mixtures $\ddot{\mu}$ and $\ddot{\nu}$. Thus, $(\ddot{\alpha}, \ddot{\beta}) \in A^+(\ddot{\mu}, \ddot{\nu}, \Pi^{\text{all}})$. It follows that $\ddot{\alpha}^+(= a_{\ddot{\mu}}^{\ddot{\nu}})$ and $\ddot{\beta}^+(= a_{\ddot{\nu}}^{\ddot{\mu}})$ are both strictly greater than 0 because $\ddot{\alpha}^+ = 0$ implies $\ddot{\alpha} = 0$ and $\ddot{\beta}^+ = 0$ implies $\ddot{\beta} = 1$ (using statement 3), which contradicts our assumption. Now, $(0, 1) \in A^+(\ddot{\mu}, \ddot{\nu}, \Pi^{\text{all}})$ follows trivially from statement 3. It is easy to observe that $(\alpha, 1) \in A^+(\ddot{\mu}, \ddot{\nu}, \Pi^{\text{all}})$ for any $\alpha \in [0, \ddot{\alpha}^+]$. Thus, there are multiple values for $\alpha$ that generate $\ddot{\mu}$ and $\ddot{\nu}$. Similarly, $(0, \beta) \in A^+(\ddot{\mu}, \ddot{\nu}, \Pi^{\text{all}})$ for any $\beta \in [1 - \ddot{\beta}^+, 1]$ and there are multiple values for $\beta$ that generate $\ddot{\mu}$ and $\ddot{\nu}$.

**Statement 4:** Statement 4 follows trivially from statement 5 and also by observing that $A^+(\ddot{\mu}, \ddot{\nu}, \Pi^{\text{all}})$ cannot be a singleton set because $\ddot{\alpha}^+, \ddot{\beta}^+ > 0$. $\blacksquare$

**Theorem 1 (Identifiablity)** *Given* $(\mu, \nu) \in \mathcal{F}(\Pi^{all})$, *let* $\alpha^+ = a_\mu^\nu$ *and* $\beta^+ = a_\nu^\mu$. *Let* $\mu_0^* = (\mu - \alpha^+ \nu)/(1 - \alpha^+)$, $\mu_1^* = (\nu - \beta^+ \mu)/(1 - \beta^+)$ *and*

$$\alpha^* = \alpha^+(1 - \beta^+)/(1 - \alpha^+ \beta^+), \qquad \beta^* = (1 - \beta^+)/(1 - \alpha^+ \beta^+). \tag{7}$$

*It follows that*

1. $(\alpha^*, \beta^*, \mu_0^*, \mu_1^*)$ *generate* $(\mu, \nu)$

2. $(\mu_0^*, \mu_1^*) \in \Pi^{res}$ *and, consequently,* $\alpha^* = a_\mu^{\mu_1^*}$, $\beta^* = a_\nu^{\mu_1^*}$.

3. $\mathcal{F}(\Pi^{res})$ *contains all pairs of mixtures in* $\mathcal{F}(\Pi^{all})$.

4. $A^+(\mu, \nu, \Pi^{res}) = \{(\alpha^*, \beta^*)\}$.

5. $\mathcal{F}(\Pi^{res})$ *is identifiable in* $(\alpha, \beta)$; *i.e.,* $(\alpha, \beta)$ *is uniquely determined from* $(\mu, \nu)$.

**Proof:**
**Statement 1:** First, we show that $\mu_0^*, \mu_1^*$ are well defined probability measures.
$\boldsymbol{\alpha^+ \neq 1, \beta^+ \neq 1}$: Suppose $\alpha^+ = 1$. It follows that $\mu = \nu$ and, consequently, from Equation 6, $\mu_0 = \mu_1 = \nu$. However, $\mu_0 \neq \mu_1$ because they are picked from $\Pi^{\text{all}}$. Thus, $\alpha^+ \neq 1$ by contradiction. Similarly $\beta^+ \neq 1$. Thus, the denominator in the R.H.S. of $\mu_0^*$ and $\mu_1^*$ is not 0.
By definition $\alpha^+ = \inf R(\mu, \nu)$. Thus, $\alpha^+ \leq \mu(A)/\nu(A)$ when $\nu(A) > 0$. Consequently, $\mu(A) - \alpha^+ \nu(A) \geq 0$. The inequality is trivially true when $\nu(A) = 0$. Thus, $\mu(A) - \alpha^+ \nu(A) \geq 0$ for all $A \in \mathcal{A}$. Hence, $\mu_0^*$ is a probability measure. Similarly, $\mu_1^*$ is also a probability measure.
Second, we show that $(\alpha^*, \beta^*, \mu_0^*, \mu_1^*)$ generate $\mu, \nu$.
$\boldsymbol{(\alpha^*, \beta^*, \mu_0^*, \mu_1^*) \to (\mu, \nu)}$: Observe that $\mu_0^*, \mu_1^*$ can also be expressed as $\mu_0^* = \frac{\beta^* \mu - \alpha^* \nu}{\beta^* - \alpha^*}$ and $\mu_1^* = \frac{(1 - \alpha^*)\nu - (1 - \beta^*)\mu}{\beta^* - \alpha^*}$. Moreover, after some algebraic manipulation of Equations labeled 7, $\alpha^*/\beta^* = \alpha^+$ and $(1 - \beta^*)/(1 - \alpha^*) = \beta^+$ can be derived. Thus, from Lemma 1 (statements 1 and 3), $(\alpha^*, \beta^*, \mu_0^*, \mu_1^*)$ generate $\mu, \nu$.

**Statement 2:** We show that $a_{\mu_1^*}^{\mu_0^*} = 0$ and $a_{\mu_0^*}^{\mu_1^*} = 0$, giving $(\mu_0^*, \mu_1^*) \in \Pi^{\text{res}}$.
We start by showing that $a_{\mu_1^*}^{\mu_0^*} = 0$. Suppose, for some $\epsilon > 0$ and

$$\nu(A) - \beta^+ \mu(A) \geq \epsilon(\mu(A) - \alpha^+ \nu(A)) \qquad \text{(for all } A \in \mathcal{A}\text{)}$$
$$\Rightarrow (1 + \epsilon\alpha^+)\nu(A) \geq (\epsilon + \beta^+)\mu(A)$$
$$\Rightarrow \nu(A)/\mu(A) \geq (\epsilon + \beta^+)/(1 + \epsilon\alpha^+) \qquad \text{(when } \mu(A) > 0\text{)}$$

Thus, $(\epsilon + \beta^+)/(1 + \epsilon\alpha^+)$ is a lower bound to $R(\nu, \mu)$ and, consequently, $\beta^+ \geq (\epsilon + \beta^+)/(1 + \epsilon\alpha^+)$. However, because $\alpha^+\beta^+ < 1$, $(\epsilon + \beta^+)/(1 + \epsilon\alpha^+) > (\epsilon\alpha^+\beta^+ + \beta^+)/(1 + \epsilon\alpha^+) = \beta^+$. This is a contradiction. Thus, for all $\epsilon > 0$ there exists some $A_\epsilon \in \mathcal{A}$ such that $\nu(A_\epsilon) - \beta^+ \mu(A_\epsilon) < \epsilon(\mu(A_\epsilon) - \alpha^+ \nu(A_\epsilon))$. We now divide the inequality on both sides by $\mu(A_\epsilon) - \alpha^+ \nu(A_\epsilon)$, observing that the divisor is strictly greater than 0 because $\nu(A_\epsilon) - \beta^+ \mu(A_\epsilon) \geq 0$ as $\mu_1^*$ is a probability measure. Thus,

$$(\nu(A_\epsilon) - \beta^+ \mu(A_\epsilon))/(\mu(A_\epsilon) - \alpha^+ \nu(A_\epsilon)) < \epsilon$$
$$\Rightarrow \mu_1^*(A_\epsilon)/\mu_0^*(A_\epsilon) < \epsilon(1 - \alpha^+)/(1 - \beta^+).$$

Because the choice of $\epsilon$ is arbitrary and $\beta^+ \neq 1$, any lower bound of $R(\mu_1^*, \mu_0^*)$ cannot be greater than 0. Thus, $a_{\mu_1^*}^{\mu_0^*} = 0$.

A similar argument shows that $a_{\mu_0^*}^{\mu_1^*} = 0$. Therefore, $a_{\mu_1^*}^{\mu_0^*} = 0$ and $a_{\mu_0^*}^{\mu_1^*} = 0$; consequently, $(\mu_0^*, \mu_1^*) \in \Pi^{\mathrm{res}}$. Because $\mu_0^*$ is not a mixture containing $\mu_1^*$ and because $\mu = \alpha^*\mu_1^* + (1-\alpha^*)\mu_0^*$, from Statement 2 in Lemma A.1 we get that $\alpha^* = a_\mu^{\mu_1^*}$. Similarly, $\beta^* = a_\nu^{\mu_1^*}$.

**Statement 3:** Because statements 1 and 2 are true for any $(\mu, \nu) \in \mathcal{F}(\Pi^{\mathrm{all}})$, statement 3 is true.

**Statement 4:** It follows from statements 1 and 2, that $(\alpha^*, \beta^*) \in A^+(\mu, \nu, \Pi^{\mathrm{res}})$. To show that $A^+(\mu, \nu, \Pi^{\mathrm{res}})$ contains no other element, we give a proof by contradiction. Suppose $(\alpha, \beta) \in A^+(\mu, \nu, \Pi^{\mathrm{res}})$ and $(\alpha, \beta) \neq (\alpha^*, \beta^*)$. Let $(\alpha, \beta, \mu_0, \mu_1)$ generate $(\mu, \nu)$, for some $(\mu_0, \mu_1) \in \Pi^{\mathrm{res}}$. First, we show that $\alpha^+ = \alpha/\beta$ and $\beta^+ = (1-\beta)/(1-\alpha)$:
$\boldsymbol{\alpha^+ = \alpha/\beta}$: Suppose for some $0 < \epsilon < (\beta-\alpha)/\beta(1-\beta)$ and all $A \in \mathcal{A}$ where $\nu(A) > 0$,

$$
\mu(A)/\nu(A) \geq \alpha/\beta + \epsilon
$$
$$
\Rightarrow \frac{\alpha + (1-\alpha)\mu_0(A)/\mu_1(A)}{\beta + (1-\beta)\mu_0(A)/\mu_1(A)} \geq \frac{\alpha + \beta\epsilon}{\beta}
$$
$$
\Rightarrow \mu_0(A)/\mu_1(A) \geq \beta^2\epsilon/(\beta-\alpha-\epsilon\beta(1-\beta))
$$
$$
\Rightarrow a_{\mu_0}^{\mu_1} \geq \beta^2\epsilon/(\beta-\alpha-\epsilon\beta(1-\beta)) > 0
$$

However, this is a contradiction because $(\mu_0, \mu_1) \in \Pi^{\mathrm{res}}$. Thus, for every $0 < \epsilon < (\beta-\alpha)/\beta(1-\beta)$ there exists some $A_\epsilon \in \mathcal{A}$ with $\nu(A_\epsilon) > 0$ such that $\mu(A_\epsilon)/\nu(A_\epsilon) < \alpha/\beta + \epsilon$. Thus, $\alpha^+ < \alpha/\beta + \epsilon$, because $\epsilon$ can be made arbitrarily small, $\alpha^+ \leq \alpha/\beta$. However, because $A^+(\mu, \nu, \Pi^{\mathrm{res}}) \subseteq A^+(\mu, \nu, \Pi^{\mathrm{all}})$, $(\alpha, \beta)$ also belongs to $A^+(\mu, \nu, \Pi^{\mathrm{all}})$ and consequently $\alpha^+ \geq \alpha/\beta$ from Lemma 1 (statement 3). Thus, $\alpha^+ = \alpha/\beta$.
$\boldsymbol{\beta^+ = (1-\beta)/(1-\alpha)}$: The proof is similar to $\alpha^+ = \alpha/\beta$ —supposing $\nu(A)/\mu(A) \geq (1-\beta)/(1-\alpha) + \epsilon$ for some $\epsilon > 0$ and all $A \in \mathcal{A}$ with $\mu(A) > 0$, reaching a contradiction and following the subsequent steps.
$\alpha^+ = \alpha/\beta$, $\beta^+ = (1-\beta)/(1-\alpha)$ and Equation 7 implies $\alpha = \alpha^*$ and $\beta = \beta^*$, which contradicts our assumption. Hence, $(\alpha^*, \beta^*)$ is the only element in $A^+(\mu, \nu, \Pi^{\mathrm{res}})$, which proves statement 4.

**Statement 5:** This statement follows by observing that $A^+(\mu, \nu, \Pi^{\mathrm{res}})$ is a singleton set. ∎

## B Proofs for properties of the univariate transform

We first prove Lemma 2, which we can then use to construct a suitable univariate transform.

**Lemma 2** *Let $X$, $Y$ and $S$ be random variables taking values in $\mathcal{X}$, $\mathcal{Y}$ and $\mathcal{S}$, respectively, and $X^u = X|S = 0$ and $X^l = X|S = 1$. For measures $\mu, \nu, \mu_0, \mu_1$, satisfying Equations 3 and 4 and $\mu_1 \neq \mu_0$, let $\mu, \mu_0, \mu_1$ give the distribution of $X$, $X|Y = 0$ and $X|Y = 1$, respectively. If $X, Y$ and $S$ satisfy assumptions 8, 9 and 10, then*

1. *$X$ is independent of $S = 0$; i.e., $p(x|S = 0) = p(x)$*
2. *$X^u$ and $X^l$ are distributed according to $\mu$ and $\nu$, respectively.*
3. *$p(S = 2) \neq 0$.*

**Proof:**

**Statement 1:** Observe that

$$
\begin{aligned}
p(x|S = 0) &= p(x, Y = 1|S = 0) + p(x, Y = 0|S = 0) \\
&= p(Y = 1|S = 0)p(x|Y = 1, S = 0) + p(Y = 0|S = 0)p(x|Y = 0, S = 0) \\
&= p(Y = 1)p(x|Y = 1) + p(Y = 0)p(x|Y = 0) \quad \text{(from assumptions 8 and 10)} \\
&= p(x).
\end{aligned}
$$

Thus, $X$ is independent of $S = 0$.

**Statement 2:** From statement 1, $X^u$ has the same distribution as $X$, which is $\mu$. Now,

$$
\begin{aligned}
p(x|S = 1) &= p(x, Y = 1|S = 1) + p(x, Y = 0|S = 1) \\
&= p(Y = 1|S = 1)p(x|Y = 1, S = 1) + p(Y = 0|S = 1)p(x|Y = 0, S = 1) \\
&= \beta p(x|Y = 1) + (1 - \beta)p(x|Y = 0). \qquad \text{(from assumptions 9 and 10)}
\end{aligned}
$$

Thus, the distribution of $X|S = 1$ is $\beta\mu_1 + (1 - \beta)\mu_0$, which is $\nu$.

**Statement 3:** Now,

$$
\begin{aligned}
p(S = 2|x) &= 1 - p(S = 0|x) - p(S = 1|x) \\
&= 1 - p(S = 0) - \frac{p(x|S = 1)}{p(x)}p(S = 1) \quad \text{(because } S = 0 \text{ and } X \text{ are independent)}
\end{aligned}
$$

The probability $p(S = 2|x)$ is independent of $x$ only if $p(x|S=1)/p(x)$ is a constant with respect to $x$. Let $p(x|S=1)/p(x) = c$, where $c$ is some constant. Integrating over $x$ on both sides gives $\int_{\mathcal{X}} p(x|S = 1)dx = c \int_{\mathcal{X}} p(x)dx$. Since both integrals are 1, it follows that $c = 1$. Thus, $p(x|S=1)/p(x) = 1$, which implies $\mu = \nu$; i.e., the labeled and unlabeled samples have the same distribution. However, this implies $\mu_1 = \mu_0$, which contradicts the assumption. Therefore, $S = 2$ is not independent of $X$. ∎

To prove the main result about the preservation properties of the univariate transform, we will make use of the following theorem.

**Lemma B.1 (Restatement of Theorem 9 in Jain et al. (2016))** *Let $X$ and $X_1$ be random variables with densities $f$ and $f_1$ and measures $\mu$ and $\mu_1$ respectively. For $\overline{\mathbb{R}}^+ = \mathbb{R}^+ \cup \{0, \infty\}$ and an abstract space $\mathcal{X}_\tau$, given any one-to-one function $G : \overline{\mathbb{R}}^+ \to \mathcal{X}_\tau$, define function $\tau : \mathcal{X} \to \mathcal{X}_\tau$*

$$
\tau = G \circ \tau_d, \qquad\qquad
\left|
\begin{array}{l}
\textit{where} \\[4pt]
\tau_d(x) = \begin{cases} f(x)/f_1(x) & \textit{if } f_1(x) > 0 \\ \infty & \textit{if } f_1(x) = 0. \end{cases}
\end{array}
\right.
$$

*Let $\mu_\tau$ and $\mu_{\tau 1}$ be the measures for the random variables $\tau(X)$, $\tau(X_1)$ respectively for $\sigma$-algebra $\mathcal{A}_\tau$ on $\mathcal{X}_\tau$. Then $a_\mu^{\mu_1} = a_{\mu_\tau}^{\mu_{\tau 1}}$.*

**Theorem 2 ($\alpha^*$-preserving transform)** *Let random variables $X, Y, S, X^u, X^l$ and measures $\mu, \nu, \mu_0, \mu_1$ be as defined in Lemma 2. Let $\tau_p$ be the posterior as defined in Equation 11 and $\tau = H \circ \tau_p$, where $H$ is a 1-to-1 function on $[0, 1]$ and $\circ$ is the composition operator. Assume*

1. *$(\mu_0, \mu_1) \in \Pi^{res}$,*
2. *$X^u$ and $X^l$ are continuous with densities $f$ and $g$, respectively,*
3. *$\mu_\tau, \nu_\tau, \mu_{\tau 1}$ are the measures corresponding to $\tau(X^u), \tau(X^l), \tau(X_1)$, respectively,*
4. *$(\alpha^+, \beta^+, \alpha^*, \beta^*) = (a_\mu^\nu, a_\nu^\mu, a_\mu^{\mu_1}, a_\nu^{\mu_1})$ and $(\alpha_\tau^+, \beta_\tau^+, \alpha_\tau^*, \beta_\tau^*) = (a_{\mu_\tau}^{\nu_\tau}, a_{\nu_\tau}^{\mu_\tau}, a_{\mu_\tau}^{\mu_{\tau 1}}, a_{\nu_\tau}^{\mu_{\tau 1}})$.*

*Then*

$$
(\alpha_\tau^+, \beta_\tau^+, \alpha_\tau^*, \beta_\tau^*) = (\alpha^+, \beta^+, \alpha^*, \beta^*)
$$

*and so $\tau$ is an $\alpha^*$-preserving transformation.*

*Moreover, $\tau_p$ can also be used to compute the true posterior probability:*

$$
p(Y = 1|x) = \frac{\alpha^*(1 - \alpha^*)}{\beta^* - \alpha^*} \left( \frac{p(S = 0)}{p(S = 1)} \frac{\tau_p(x)}{1 - \tau_p(x)} - \frac{1 - \beta^*}{1 - \alpha^*} \right). \tag{12}
$$

**Proof:** First we prove that $(\alpha^+, \beta^+) = (\alpha_\tau^+, \beta_\tau^+)$. To this end, we expand $\tau_p(x)$ as follows

$$\tau_p(x) = \frac{p(S = 1, x, S \in \{0, 1\})}{p(x, S \in \{0, 1\})}$$

$$= \frac{p(S = 1, x)}{p(x, S = 0) + p(x, S = 1)}$$

$$= \frac{p(x|S = 1)p(S = 1)}{p(x|S = 0)p(S = 0) + p(x|S = 1)p(S = 1)} \tag{14}$$

$$= \frac{p(S = 1)}{\frac{p(x|S=0)}{p(x|S=1)}p(S = 0) + p(S = 1)}$$

$$= \frac{p(S = 1)}{\frac{f(x)}{g(x)}p(S = 0) + p(S = 1)} \tag{15}$$

the last step is justified because $f(x) = p(X = x|S = 0)$ and $g(x) = p(X = x|S = 1)$.

Consider one-to-one functions $G_1(t) = \frac{p(S=1)}{tp(S=0)+p(S=1)}$ and $G_2(t) = \frac{p(S=1)}{\frac{1}{t}p(S=0)+p(S=1)}$ defined on $\mathbb{R}^+ \cup \{0, \infty\}$. We can apply Lemma B.1 to $G_1$ to get $\alpha^+ = \alpha_\tau^+$ and to $G_2$ to get $\beta^+ = \beta_\tau^+$. To satisfy the condition of Lemma B.1 for $G_1$, let $X^u$, $X^l$ and $H \circ G_1$ play the role of $X$, $X_1$ and $G$, respectively. Now,

$$\tau = H \circ \tau_p$$

$$= H \circ G_1 \circ \tau_d \qquad \text{(from Equation 15)}$$

Because $H$ and $G_1$ are both one-to-one functions, $H \circ G_1$ is one-to-one as well. Thus, all the conditions of Lemma B.1 are satisfied and, consequently, $\alpha^+ = \alpha_\tau^+$.

Similarly, we can use Lemma B.1 with $X^u$, $X^l$ and $H \circ G_2$ playing the role of $X_1$, $X$ and $G$, respectively, now giving that $\beta^+ = \beta_\tau^+$.

From Statement 2 in Theorem 1 and Equation 7

$$\alpha^* = \alpha^+(1-\beta^+)/(1-\alpha^+\beta^+)$$

$$\beta^* = (1-\beta^+)/(1-\alpha^+\beta^+)$$

$$\alpha_\tau^* = \alpha_\tau^+(1-\beta_\tau^+)/(1-\alpha_\tau^+\beta_\tau^+)$$

$$\beta_\tau^* = (1-\beta_\tau^+)/(1-\alpha_\tau^+\beta_\tau^+).$$

and, thus, $(\alpha_\tau^*, \beta_\tau^*) = (\alpha^*, \beta^*)$.

Next we prove Equation 12. Let $c = p(S=0)/p(S=1)$. Rearranging the terms of Equation 14,

$$\frac{\tau(x)}{1 - \tau(x)} = \frac{p(x|S = 1)p(S = 1)}{p(x|S = 0)p(S = 0)}$$

$$= \frac{p(S = 1)}{p(S = 0)}\left(\frac{p(x, Y = 1|S = 1) + p(x, Y = 1|Y = 0)}{p(x)}\right) \qquad \text{(from Lemma 2 statement 1)}$$

$$= \frac{p(S = 1)}{p(S = 0)}\left(p(Y = 1|S = 1)\frac{p(x|Y = 1, S = 1)}{p(x)} + p(Y = 0|S = 1)\frac{p(x|Y = 0, S = 1))}{p(x)}\right)$$

$$= \frac{p(S = 1)}{p(S = 0)}\left(p(Y = 1|S = 1)\frac{p(x|Y = 1)}{p(x)} + p(Y = 0|S = 1)\frac{p(x|Y = 0)}{p(x)}\right)$$

$$\text{(from assumption 10)}$$

$$= \frac{1}{c}\left(\frac{p(Y = 1|S = 1)}{p(Y = 1)}p(Y = 1|x) + \frac{p(Y = 0|S = 1)}{p(Y = 0)}p(Y = 0|x)\right)$$

$$= \frac{1}{c}\left(\frac{\beta^*}{\alpha^*}p(Y = 1|x) + \frac{1 - \beta^*}{1 - \alpha^*}(1 - p(Y = 1|x))\right)$$

$$= \frac{1}{c}\left(\frac{1 - \beta^*}{1 - \alpha^*} + \left(\frac{\beta^*}{\alpha^*} - \frac{1 - \beta^*}{1 - \alpha^*}\right)p(Y = 1|x)\right).$$

Rearranging the terms,

$$p(Y = 1|x) = \frac{\alpha^*(1-\alpha^*)}{\beta^* - \alpha^*}\left(\frac{c\tau(x)}{1-\tau(x)} - \frac{1-\beta^*}{1-\alpha^*}\right).$$

$\blacksquare$

## C   MSGMM

Let $U = \{X_i^u\}$ and $L = \{X_i^l\}$ be the unlabeled sample and the noisy positive sample, respectively. The parametric approach is derived by modeling each sample as a two component Gaussian mixture, sharing the same components but having different mixing proportions:

$$X_i^u \sim \alpha\mathcal{N}(u_1, \Sigma_1) + (1-\alpha)\mathcal{N}(u_0, \Sigma_0)$$
$$X_i^l \sim \beta\mathcal{N}(u_1, \Sigma_1) + (1-\beta)\mathcal{N}(u_0, \Sigma_0)$$

where $u_1, u_0 \in \mathbb{R}^d$ and $\Sigma_1, \Sigma_0 \in \mathbb{S}_{++}^d$, the set of all $d \times d$ positive definite matrices. The algorithm is an extension to the EM approach for Gaussian mixture models (GMMs) where, instead of estimating the parameters of a single mixture, the parameters of both mixtures $(\alpha, \beta, u_0, u_1, \Sigma_0, \Sigma_1)$ are estimated simultaneously by maximizing the combined likelihood over both $U$ and $L$. This approach, that we refer to as a multi-sample GMM (MSGMM), exploits the constraint that the two mixtures share the same components.

To derive the update equations, we introduce missing variables $W_i^u, W_j^l$ that give the true class of the $i^{th}$ and $j^{th}$ example in $U$ and $L$, respectively. The variables $W_i^u, W_j^l$ are Bernoulli distributed; i.e., $W_i^u \sim \text{Bernoulli}(\alpha)$ and $W_j^l \sim \text{Bernoulli}(\beta)$. For

$$W = \{W_i^u\}_{i=1}^{|U|}, V = \{W_j^l\}_{j=1}^{|L|}$$

the quartet $(U, L, W, V)$ forms the observed and unobserved variables in the EM framework. The complete data log-likelihood, $ll_C$ is given by,

$$ll_C = \sum_{i=1}^{|U|} W_i^u \log[\alpha\phi_1(x_i^u)] + (1-W_i^u)\log[(1-\alpha)\phi_0(x_i^u)]$$
$$+ \sum_{i=1}^{|L|} W_i^l \log[\beta\phi_1(x_i^l)] + (1-W_i^l)\log[(1-\beta)\phi_0(x_i^l)],$$

where $\phi_i$ is the density of $\mathcal{N}(u_i, \Sigma_i)$. Our goal is to maximize $E[ll_C]$. To do so we take the conditional expectation of $ll_C$ with respect to $W$ and $V$ given $U$ and $L$. For

$$\bar{w}_i^u = E[W_i^u|X_i^u = x_i^u] = \frac{\alpha\phi_1(x_i^u)}{\alpha\phi_1(x_i^u) + (1-\alpha)\phi_0(x_i^u)},$$

$$\bar{w}_i^l = E[W_i^l|X_i^l = x_i^l] = \frac{\beta\phi_1(x_i^l)}{\beta\phi_1(x_i^l) + (1-\beta)\phi_0(x_i^l)},$$

we obtain

$$E[ll_C] = \sum_{i=1}^{|U|} \bar{w}_i^u \log[\alpha\phi_1(x_i^u)] + (1 - \bar{w}_i^u)\log[(1-\alpha)\phi_0(x_i^u)]$$

$$+ \sum_{i=1}^{|L|} \bar{w}_i^l \log[\beta\phi_1(x_i^l)] + (1-\bar{w}_i^l)\log[(1-\beta)\phi_0(x_i^l)]$$

which up to constants, that are ignored in the optimization, can be explicitly written as

$$E[ll_C] = \sum_{i=1}^{|U|} \bar{w}_i^u\left[\log\alpha - \frac{1}{2}\log|\Sigma_1| - (x_i^u - u_1)^T\Sigma_1^{-1}(x_i^u - u_1)\right]$$

$$+ \sum_{i=1}^{|U|}(1-\bar{w}_i^u)\left[\log(1-\alpha) - \frac{1}{2}\log|\Sigma_0| - (x_i^u - u_0)^T\Sigma_0^{-1}(x_i^u - u_0)\right]$$

$$+ \sum_{i=1}^{|L|} \bar{w}_i^l\left[\log\beta - \frac{1}{2}\log|\Sigma_1| - (x_i^l - u_1)^T\Sigma_1^{-1}(x_i^l - u_1)\right]$$

$$+ \sum_{i=1}^{|L|}(1-\bar{w}_i^l)\left[\log(1-\beta) - \frac{1}{2}\log|\Sigma_0| - (x_i^l - u_0)^T\Sigma_0^{-1}(x_i^l - u_0)\right].$$

Finally, we obtain the parameter update equations by maximizing $E[ll_C]$ with respect to $(\alpha, \beta, u_0, u_1, \Sigma_0, \Sigma_1)$:

$$\alpha \leftarrow {}^1\!/\!|U| \sum_{i=1}^{|U|} \bar{w}_i^u$$

$$\beta \leftarrow {}^1\!/\!|L| \sum_{j=1}^{|L|} \bar{w}_j^l$$

$$u_1 \leftarrow \frac{\sum_{i=1}^{|U|} \bar{w}_i^u x_i^u + \sum_{j=1}^{|L|} \bar{w}_j^l x_j^l}{\sum_{i=1}^{|U|} \bar{w}_i^u + \sum_{j=1}^{|L|} \bar{w}_j^l}$$

$$u_0 \leftarrow \frac{\sum_{i=1}^{|U|} (1 - \bar{w}_i^u) x_i^u + \sum_{j=1}^{|L|} (1 - \bar{w}_j^l) x_j^l}{\sum_{i=1}^{|U|} (1 - \bar{w}_i^u) + \sum_{j=1}^{|L|} (1 - \bar{w}_j^l)}$$

$$\Sigma_1 \leftarrow \frac{\sum_{i=1}^{|U|} \bar{w}_i^u (x_i^u - u_1)(x_i^u - u_1)^T + \sum_{j=1}^{|L|} \bar{w}_j^l (x_j^l - u_1)(x_j^l - u_1)^T}{\sum_{i=1}^{|U|} \bar{w}_i^u + \sum_{j=1}^{|L|} \bar{w}_j^l}$$

$$\Sigma_0 \leftarrow \frac{\sum_{i=1}^{|U|} (1 - \bar{w}_i^u)(x_i^u - u_0)(x_i^u - u_0)^T + \sum_{j=1}^{|L|} (1 - \bar{w}_j^l)(x_j^l - u_0)(x_j^l - u_0)^T}{\sum_{i=1}^{|U|} (1 - \bar{w}_i^u) + \sum_{j=1}^{|L|} (1 - \bar{w}_j^l)}$$

The update rules reduce to the standard GMM when the labeled sample is not provided. Further generalization to more than two samples and/or mixing components is straightforward.

## D   AlphaMax

For a mixture sample $M$ and a component sample $C$, AlphaMax$(M, C)$ estimates the maximum proportion of $C$ in $M$ (Jain et al., 2016). AlphaMax is based on the constrained maximization of the log likelihood of samples $M$ and $C$, derived using nonparametric estimates of their densities $m$ and $c$, respectively. We list the main steps of AlphaMax below.

1. Estimate $c$ nonparameterically as $\hat{c}$ using sample $C$. Obtain the weights, $v_i$, and components $\kappa_i$ from nonparametric density estimation of $m$ as a $k$-component mixture, $\hat{m}(x) = \sum_{i=1}^{k} v_i \kappa_i(x)$, using $M$.

2. Construct two density functions $\tilde{c}(\cdot | \boldsymbol{\omega})$ and $\tilde{m}(\cdot | \boldsymbol{\omega})$ from $v_i, \kappa_i$ and $\hat{c}$ parameterized by a $k$-dimensional weight vector $\boldsymbol{\omega} = [\omega_i], 0 \leq \omega_i \leq 1$, which re-weights components $\kappa_i$:

$$\tilde{c}(x | \boldsymbol{\omega}) = \frac{\sum_{i=1}^{k} \omega_i v_i \kappa_i(x)}{\sum_{i=1}^{k} \omega_i v_i},$$

$$\tilde{m}(x | \boldsymbol{\omega}) = \left( \sum_{i=1}^{k} \omega_i v_i \right) \hat{c}(x) + \left( 1 - \sum_{i=1}^{k} \omega_i v_i \right) \tilde{c}(x | 1 - \boldsymbol{\omega});$$

3. Maximize the log likelihood of $M$ and $C$ constructed with $\tilde{m}$ and $\tilde{c}$ under the constraint $\sum_{i=1}^{k} \omega_i v_i = r$ for many values of $r$ equispaced in $[0, 1]$.

$$ll_r = \max_{w.r.t.\ \boldsymbol{\omega}} \sum_{x \in M} \log \tilde{m}(x | \boldsymbol{\omega}) + \sum_{x \in C} \log \tilde{c}(x | \boldsymbol{\omega}),$$

$$\text{subject to} \quad \begin{array}{l} \sum_{i=1}^{k} \omega_i v_i = r, \\ 0 \leq \omega_i \leq 1, i = 1, \dots, k. \end{array}$$

4. Estimate the maximum proportion of $c$ in $m$, $a_m^c$ (minor abuse of notation[1]), as the $x$-coordinate of the elbow in the $ll_r$ versus $r$ graph.

The densities $\tilde{m}(\cdot | \boldsymbol{\omega})$ and $\tilde{c}(\cdot | \boldsymbol{\omega})$ are constructed to approximate $m$ and $c$. The efficacy of the approximation depends on the value of $\boldsymbol{\omega}$; there exists $\boldsymbol{\omega}$ such that $\tilde{m}(\cdot | \boldsymbol{\omega})$ and $\tilde{c}(\cdot | \boldsymbol{\omega})$ are good approximations provided $\sum_{i=1}^{k} \omega_i v_i \leq a_m^c$, however, the approximation deteriorates progressively, even

with the optimum $\boldsymbol{\omega}$, as $\sum_{i=1}^{k} \omega_i v_i$ moves beyond $a_m^c$. This suggests that the graph of $ll_r$ versus $r$ should be approximately a flat line from 0 to $a_m^c$ and decrease progressively beyond $a_m^c$ exposing an elbow at $a_m^c$, which is detected in the last step. The pseudo code for elbow detection is provided in (Jain et al., 2016).

The practical implementation, backed by the $\alpha^*$-preservation theory, reduces the dimension of the data to a single dimension by using the scoring function of a non-traditional classifier and employs histograms as the nonparametric method to obtain $\hat{m}$ and $\hat{c}$. The bin-width is chosen to cover the component sample's (after the transformation) range and reveal the shape of its distribution, using the default option in Matlab's `histogram` function. More bins with the same bin-width are subsequently added to cover the mixture sample's range.

The total computation includes the time to (a) train a classifier (b) perform density estimation in 1D and (c) perform optimization in AlphaMax. The complexity for (a) varies; for neural networks it is $O(n)$ per epoch. For (b) we used $k$ bin histograms, where $k$ can be $O(\log(n))$ to $O(\sqrt{n})$ depending on the bandwidth selection rule, giving $O(nk)$ complexity. For (c), (size $k$ optimization) the computation of the objective and gradient is $O(nk)$ per step; e.g., LBFGS is $O(nk)$ per step. An execution of AlphaMax takes about 10 minutes on a laptop computer for the Shuttle data set (for results reported in Table 1; i.e., 1000 labeled and 10000 unlabeled examples); 12 hours on the entire data (8903 labeled and 49097 unlabeled examples).

## E Empirical results

**Results for univariate synthetic data.** The results for synthetic data, with scalar inputs, are summarized in Table 2.

**Results for multivariate AlphaMax-N and MSGMM.** To demonstrate the efficacy of the class prior preserving transform, we implemented the multivariate versions of AlphaMax-N and MS-GMM and evaluated them on the twelve real data sets without applying the transform. There were significant stability and computational issues related to the high-dimensional nature of the data sets. MSGMM was numerically unstable because of singular/nearly-singular covariance matrix, whereas AlphaMax-N became computationally demanding because the number of bins (for histogram based density estimation) grow exponentially with the dimension, resulting in a large parameter vector $\boldsymbol{\omega}$ and, consequently, a large optimization problem, even after removing the zero-count bins. This is expected, as density estimation for multivariate data is known to be problematic, which is one of the main reasons for introducing our transform. To make estimation feasible under these stability and computational issues, we used dimensionality reduction. Though not all data sets posed the same level of difficulty, to have a standard approach and permit effective density estimation, we used the top three principal components, obtained via principal component analysis on the z-score normalized data (mixture and component samples combined), as input to the two algorithms. We also attempted using top $k$ principal components that preserve 75 percent of the total variance, however, for some of the data sets, the dimension was still too high.

In the same manner as in the univariate case, we used histograms in the multivariate implementation of AlphaMax-N. The bin-width for a dimension was selected to minimize the asymptotic mean integrated squared error (AMISE) with a normal reference rule, using the component sample, $C$. The formula for the bin-width of dimension $k$ is given by:

$$b_k = 3.5\sigma_k |C|^{-1/(2+d)},$$

where $d$ is the total number of dimensions, $\sigma_k$ is the standard deviation of the $k^{th}$ dimension and $|C|$ is the size of the component sample. Bins were added to cover the range of the entire data, mixture and component combined, and empty bins were removed to reduce the size of the optimization problem.

Table 3 contains the results of AlphaMax-N and MSGMM on the real-life data sets, using the top three principal components under column headings AlphaMax-NM (M for multi-dimensional) and MSGMM, respectively. The results of AlphaMax-N and MSGMM-T with the class-prior preserving transform are also provided for comparison. Notice that, though AlphaMax-NM (without transform) performs well, AlphaMax-N (with transform) is significantly better in terms of estimation error, despite having a lower computational cost. Also notice the deterioration in the performance of MSGMM (without transform) compared to MSGMM-T (with transform).

Table 2: Mean absolute difference between estimated and true mixing proportion over a selection of true mixing proportions and the following data sets: $\mathcal{N}$ = Gaussian with $\Delta\mu \in \{1, 2, 4\}$, $\mathcal{L}$ = Laplace with $\Delta\mu \in \{1, 2, 4\}$. Statistical significance was evaluated by comparing the Elkan-Noto algorithm, AlphaMax, AlphaMax-N, and the multi-sample Gaussian Mixture Model (MSGMM). The bold font type indicates the winner and the asterisk indicates statistical significance.

| Data | $\alpha$ | $\beta$ | Elkan-Noto 100 | Elkan-Noto 1000 | AlphaMax 100 | AlphaMax 1000 | AlphaMax-N 100 | AlphaMax-N 1000 | MSGMM 100 | MSGMM 1000 |
|---|---|---|---|---|---|---|---|---|---|---|
| $\mathcal{N}$ $\Delta\mu=1$ | 0.05 | 1.00 | 0.473 | 0.460 | 0.079 | 0.102 | 0.064 | 0.085 | **0.034*** | **0.020*** |
| | 0.25 | 1.00 | 0.484 | 0.435 | 0.132 | 0.160 | 0.123 | 0.122 | **0.063*** | **0.044*** |
| | 0.50 | 1.00 | 0.395 | 0.347 | 0.155 | 0.125 | 0.173 | 0.096 | **0.073*** | **0.040*** |
| | 0.05 | 0.95 | 0.484 | 0.496 | 0.099 | 0.124 | 0.076 | 0.099 | **0.039*** | **0.022*** |
| | 0.25 | 0.95 | 0.510 | 0.469 | 0.127 | 0.167 | 0.115 | 0.111 | **0.074*** | **0.037*** |
| | 0.50 | 0.95 | 0.433 | 0.378 | 0.180 | 0.152 | 0.186 | 0.087 | **0.068*** | **0.048*** |
| | 0.05 | 0.75 | 0.663 | 0.630 | 0.124 | 0.135 | 0.089 | 0.076 | **0.056*** | **0.024*** |
| | 0.25 | 0.75 | 0.641 | 0.608 | 0.152 | 0.209 | 0.141 | 0.120 | **0.141** | **0.060*** |
| | 0.50 | 0.75 | 0.548 | 0.485 | 0.219 | 0.218 | 0.244 | 0.137 | **0.040*** | **0.068*** |
| $\mathcal{N}$ $\Delta\mu=2$ | 0.05 | 1.00 | 0.112 | 0.136 | 0.018 | 0.016 | 0.017 | 0.015 | **0.006*** | **0.004*** |
| | 0.25 | 1.00 | 0.177 | 0.168 | 0.050 | 0.049 | 0.049 | 0.042 | **0.018*** | **0.010*** |
| | 0.50 | 1.00 | 0.219 | 0.153 | 0.104 | 0.053 | 0.109 | 0.043 | **0.024*** | **0.015*** |
| | 0.05 | 0.95 | 0.125 | 0.162 | 0.015 | 0.019 | 0.015 | 0.015 | **0.009*** | **0.004*** |
| | 0.25 | 0.95 | 0.212 | 0.208 | 0.063 | 0.061 | 0.060 | 0.043 | **0.025*** | **0.011*** |
| | 0.50 | 0.95 | 0.271 | 0.211 | 0.099 | 0.077 | 0.106 | 0.046 | **0.023*** | **0.018*** |
| | 0.05 | 0.75 | 0.215 | 0.285 | 0.030 | 0.036 | **0.022** | 0.012 | 0.034 | **0.005*** |
| | 0.25 | 0.75 | 0.405 | 0.403 | 0.100 | 0.139 | 0.084 | 0.036 | **0.025*** | **0.014*** |
| | 0.50 | 0.75 | 0.457 | 0.415 | 0.156 | 0.184 | 0.159 | 0.048 | **0.030*** | **0.020*** |
| $\mathcal{N}$ $\Delta\mu=4$ | 0.05 | 1.00 | 0.010 | 0.013 | 0.022 | 0.019 | 0.022 | 0.019 | **0.001*** | **0.001*** |
| | 0.25 | 1.00 | 0.032 | 0.018 | 0.022 | 0.004 | 0.027 | 0.008 | **0.002*** | **0.002*** |
| | 0.50 | 1.00 | 0.070 | 0.020 | 0.032 | 0.005 | 0.039 | 0.012 | **0.002*** | **0.002*** |
| | 0.05 | 0.95 | 0.018 | 0.029 | 0.018 | 0.014 | 0.019 | 0.015 | **0.001*** | **0.001*** |
| | 0.25 | 0.95 | 0.063 | 0.056 | 0.010 | 0.017 | 0.013 | 0.009 | **0.002*** | **0.001*** |
| | 0.50 | 0.95 | 0.115 | 0.083 | 0.040 | 0.032 | 0.045 | 0.010 | **0.002*** | **0.001*** |
| | 0.05 | 0.75 | 0.062 | 0.114 | 0.011 | 0.006 | 0.018 | 0.015 | **0.001*** | **0.001*** |
| | 0.25 | 0.75 | 0.236 | 0.249 | 0.063 | 0.090 | 0.026 | 0.005 | **0.002*** | **0.002*** |
| | 0.50 | 0.75 | 0.380 | 0.353 | 0.130 | 0.168 | 0.106 | 0.021 | **0.002*** | **0.002*** |
| $\mathcal{L}$ $\Delta\mu=1$ | 0.05 | 1.00 | 0.410 | 0.389 | 0.195 | 0.256 | **0.147*** | **0.190*** | 0.418 | 0.390 |
| | 0.25 | 1.00 | 0.410 | 0.356 | 0.151 | 0.209 | **0.103*** | **0.117*** | 0.148 | 0.183 |
| | 0.50 | 1.00 | 0.367 | 0.299 | 0.195 | 0.154 | **0.190** | **0.038*** | 0.243 | 0.236 |
| | 0.05 | 0.95 | 0.455 | 0.430 | 0.204 | 0.271 | **0.140*** | **0.192*** | 0.424 | 0.406 |
| | 0.25 | 0.95 | 0.455 | 0.401 | 0.180 | 0.230 | **0.115*** | **0.112*** | 0.157 | 0.187 |
| | 0.50 | 0.95 | 0.412 | 0.331 | 0.222 | 0.179 | **0.214** | **0.047*** | 0.249 | 0.241 |
| | 0.05 | 0.75 | 0.593 | 0.578 | 0.225 | 0.325 | **0.133*** | **0.181*** | 0.415 | 0.440 |
| | 0.25 | 0.75 | 0.602 | 0.562 | 0.191 | 0.327 | **0.119*** | **0.123*** | 0.210 | 0.230 |
| | 0.50 | 0.75 | 0.520 | 0.470 | 0.264 | 0.278 | 0.272 | **0.053*** | **0.254** | 0.247 |
| $\mathcal{L}$ $\Delta\mu=2$ | 0.05 | 1.00 | 0.116 | 0.123 | 0.052 | 0.061 | **0.045** | 0.056 | 0.448 | **0.014*** |
| | 0.25 | 1.00 | 0.158 | 0.132 | 0.054 | 0.067 | 0.041 | 0.051 | **0.037** | **0.027*** |
| | 0.50 | 1.00 | 0.186 | 0.125 | 0.077 | 0.049 | **0.074** | **0.021*** | 0.130 | 0.068 |
| | 0.05 | 0.95 | 0.131 | 0.146 | 0.053 | 0.066 | **0.042*** | 0.053 | 0.455 | **0.016*** |
| | 0.25 | 0.95 | 0.186 | 0.175 | 0.071 | 0.082 | **0.049** | 0.049 | 0.055 | **0.021*** |
| | 0.50 | 0.95 | 0.239 | 0.183 | 0.082 | 0.082 | **0.073** | **0.024*** | 0.134 | 0.075 |
| | 0.05 | 0.75 | 0.230 | 0.268 | 0.079 | 0.097 | **0.046*** | **0.052*** | 0.461 | 0.261 |
| | 0.25 | 0.75 | 0.375 | 0.377 | 0.123 | 0.159 | **0.090*** | **0.039** | 0.191 | 0.050 |
| | 0.50 | 0.75 | 0.450 | 0.406 | 0.201 | 0.212 | 0.216 | **0.045** | **0.101*** | 0.067 |
| $\mathcal{L}$ $\Delta\mu=4$ | 0.05 | 1.00 | 0.015 | 0.020 | 0.024 | 0.020 | 0.025 | 0.021 | **0.015** | **0.011*** |
| | 0.25 | 1.00 | 0.040 | 0.025 | 0.018 | **0.005** | 0.022 | 0.005 | **0.009*** | 0.009 |
| | 0.50 | 1.00 | 0.094 | 0.027 | 0.022 | 0.005 | 0.028 | 0.006 | **0.002*** | **0.002*** |
| | 0.05 | 0.95 | 0.024 | 0.037 | **0.019** | 0.017 | 0.020 | 0.018 | 0.022 | **0.012*** |
| | 0.25 | 0.95 | 0.074 | 0.063 | 0.013 | 0.021 | 0.011 | 0.011 | **0.009** | **0.009*** |
| | 0.50 | 0.95 | 0.141 | 0.089 | 0.028 | 0.034 | 0.026 | 0.008 | **0.002*** | **0.002*** |
| | 0.05 | 0.75 | 0.072 | 0.124 | **0.008*** | **0.005*** | 0.015 | 0.010 | 0.034 | 0.012 |
| | 0.25 | 0.75 | 0.244 | 0.258 | 0.076 | 0.095 | 0.033 | 0.009 | **0.009*** | **0.008** |
| | 0.50 | 0.75 | 0.389 | 0.355 | 0.128 | 0.173 | 0.106 | 0.014 | **0.002*** | **0.002*** |

Table 3: Mean absolute difference between estimated and true mixing proportion over twelve data sets from the UCI Machine Learning Repository. Statistical significance was evaluated by comparing AlphaMax-NM, MSGMM (both using top three principal components of the data as input), AlphaMax-N, MSGMM-T (both using class-prior preserving transform). The bold font type indicates the winner and the asterisk indicates statistical significance. For each data set, shown are the true mixing proportion ($\alpha$), true proportion of the positives in the labeled sample ($\beta$) and the percent of the total variance explained by the top three principal components rounded to the nearest integer (%variance).

| Data | $\alpha$ | $\beta$ | %variance | AlphaMax-N | AlphaMax-NM | MSGMM-T | MSGMM |
|------|------|------|------|------|------|------|------|
| Bank | 0.095 | 1.00 | | 0.037 | **0.028*** | 0.163 | 0.528 |
| | 0.096 | 0.95 | 40 | 0.036 | **0.032** | 0.155 | 0.574 |
| | 0.101 | 0.75 | | **0.040** | 0.047 | 0.127 | 0.580 |
| Concrete | 0.419 | 1.00 | | 0.181 | 0.276 | 0.077 | **0.020*** |
| | 0.425 | 0.95 | 60 | 0.231 | 0.269 | 0.095 | **0.028*** |
| | 0.446 | 0.75 | | 0.272 | 0.320 | 0.233 | **0.063*** |
| Gas | 0.342 | 1.00 | | 0.017 | 0.030 | **0.008*** | 0.585 |
| | 0.353 | 0.95 | 81 | **0.006** | 0.021 | 0.006 | 0.575 |
| | 0.397 | 0.75 | | 0.009 | 0.064 | **0.006*** | 0.533 |
| Housing | 0.268 | 1.00 | | **0.094*** | 0.132 | 0.209 | 0.316 |
| | 0.281 | 0.95 | 69 | 0.110 | **0.110** | 0.204 | 0.308 |
| | 0.330 | 0.75 | | **0.134** | 0.205 | 0.172 | 0.283 |
| Landsat | 0.093 | 1.00 | | **0.007** | 0.008 | 0.157 | 0.443 |
| | 0.103 | 0.95 | 89 | **0.008*** | 0.028 | 0.152 | 0.298 |
| | 0.139 | 0.75 | | **0.012*** | 0.053 | 0.143 | 0.270 |
| Mushroom | 0.409 | 1.00 | | **0.022*** | 0.075 | 0.037 | 0.432 |
| | 0.416 | 0.95 | 24 | **0.008*** | 0.021 | 0.037 | 0.398 |
| | 0.444 | 0.75 | | **0.020** | 0.050 | 0.024 | 0.375 |
| Pageblock | 0.086 | 1.00 | | **0.044** | 0.046 | 0.129 | 0.178 |
| | 0.087 | 0.95 | 72 | 0.052 | **0.040*** | 0.125 | 0.178 |
| | 0.090 | 0.75 | | 0.064 | **0.031*** | 0.111 | 0.188 |
| Pendigit | 0.243 | 1.00 | | **0.009*** | 0.071 | 0.081 | 0.289 |
| | 0.248 | 0.95 | 65 | **0.005*** | 0.070 | 0.074 | 0.286 |
| | 0.268 | 0.75 | | **0.007*** | 0.092 | 0.062 | 0.260 |
| Pima | 0.251 | 1.00 | | **0.111** | 0.123 | 0.171 | 0.299 |
| | 0.259 | 0.95 | 60 | **0.110*** | 0.156 | 0.168 | 0.292 |
| | 0.289 | 0.75 | | **0.156** | 0.178 | 0.175 | 0.286 |
| Shuttle | 0.139 | 1.00 | | **0.029*** | 0.064 | 0.157 | 0.232 |
| | 0.140 | 0.95 | 67 | **0.007*** | 0.055 | 0.157 | 0.227 |
| | 0.143 | 0.75 | | **0.004*** | 0.015 | 0.148 | 0.356 |
| Spambase | 0.226 | 1.00 | | 0.041 | **0.034** | 0.059 | 0.487 |
| | 0.240 | 0.95 | 22 | 0.042 | **0.041** | 0.063 | 0.485 |
| | 0.295 | 0.75 | | **0.044*** | 0.072 | 0.059 | 0.434 |
| Wine | 0.566 | 1.00 | | **0.060** | 0.258 | 0.070 | 0.134 |
| | 0.575 | 0.95 | 65 | **0.063** | 0.256 | 0.076 | 0.126 |
| | 0.612 | 0.75 | | 0.353 | 0.302 | 0.293 | **0.096*** |