[Reviews · NeurIPS 2016]

Reviewer 1

Summary

The paper proposes a method for estimating the class prior given on labelled sample, as well as noisy positive samples. This builds upon an existing approach to estimate the prior when the positive samples are clean. Experiments show good performance compared to some existing methods.

Qualitative Assessment

UPDATE AFTER REBUTTAL: Regarding the main result highlighted in the rebuttal: it's indeed nice to relate the observed P(y|x) to the true one, but again this was already done in [1] using similar ideas, just without noise in the positives. The result here thus didn't really surprise me. As mentioned in my review, it's certainly valuable to have all this written down, but to me falls below the novelty threshold. Regarding the related work on asymmetric noise: my basic point is that from (Scott et al., 2013) Lemma 1 + Proposition 3 (part 2), one has a means of relating the output of a "mixture proportion estimator" (MPE) to the mixing weights for generic asymmetric positive and negative label noise. As I understand, that means that the estimator of Sanderson & Scott that I mentioned could be used in the asymmetric noise scenario. The authors are correct that Sanderson & Scott focussed on an application to (multiclass) positive + unlabelled learning, without noise in the positives; but the point remains that their estimator is conceivably applicable in the scenario of noisy positives as well. I believe this is similarly true of Ramaswamy et al.'s estimator. ============== Learning from positive and unlabelled data is a fairly well studied problem, and the paper makes a reasonable case for attention to be made in the case where the positives are subject to noise. Existing methods in the PU literature for estimating the class prior are not directly applicable here, and so it is of interest to design methods for the scenario. The paper thus is well motivated. On the technical side, the paper has a fairly careful theoretical analysis of the identifiably issues in this problem, which is appreciated. This is used to determine a canonical form in which estimation is performed, leading to a practical algorithm for the estimation. The basic idea is to build upon the AlphaMax framework of (Jain et al., 2016), and cast the estimation problem as determining the mixing weight for a certain mixture model. This estimation is performed on the outputs of any probabilistically-consistent scorer on the inputs (i.e. it does not suffer from the curse of dimensionality.) Thus, the basic technical content seems sound. I do however have two concerns about the novelty of the work. (1) The analysis in section 3 appears to closely follow that of (Jain et al., 2016), with some natural extensions to account for the fact that one is in the noisy positive setting. Unless I missed something, there don't seem to be any fundamentally new insights or techniques used in the proofs of the section: the two basic messages of (Jain et al., 2016) are translated, namely that working with all possible distributions leads to non-identifiability, while disallowing distributions that are themselves mixtures alleviates the problem. One has two calls to AlphaMax for the estimation, but the basic idea of mixture modelling is as per the original paper. The idea of then using univariate transformations in section 4 appears particularly similar to what is done in (Jain et al., 2016). The paper claims that the latter was restricted to the case where the transformation results in calibrated probability, but doesn't Thm 9 of the latter also allow for arbitrary transformations? Certainly it is valuable for all this to be written down in the noisy PU setting, but this leads to my next concern. (2) The precise distinction of the scenario considered here to the general asymmetric label noise setting, seems worthy of a little more discussion. This is first to make clear why one does not just conduct all analysis in the general case, rather than assuming that one of the observed distributions is in fact the underlying marginal (i.e. why one does not work in the general mutually contaminated framework of Scott et al., 2013). Is there some reason analogues of Theorem 1 would not exist here? (As noted, at least some of the results have precedent in Scott et al., 2013, albeit in a different language.) The second reason is there are methods for the latter which I think could be applied for the scenario considered here, in which case the paper should compare against them. In particular, I refer to the practically viable estimators of Sanderson and Scott. Class proportion estimation with application to multiclass anomaly rejection. AISTATS 2014. Liu and Tao. Classification with Noisy Labels by Importance Reweighting. PAMI 2016. (arXiV draft since 2014) Ramaswamy et al. Mixture Proportion Estimation via Kernel Embedding of Distributions. ICML 2016. (arXiV draft since early 2016, so reasonable for authors to be unaware of) All three of the above seem like they would be relevant at the end of the third paragraph in the introduction, as a contrast to the cited work of Scott et al., 2013 that relies on density estimation. I note that the first method in particular also does not require the use of calibrated probabilities, but rather a probabilistically consistent ranker (as the estimate is based on a derivative of the RHS of the ROC curve). Overall, I think the paper's contribution is reasonable, being an extension of (Jain et al., 2016) to the noisy PU case, but I think the novelty is a weak point. Other comments: - section 3 is said to include "a few missing results needed for our approach" -- explicitly identifying which these are seems prudent. - I found the use of a subscript and superscript in a^lambda_mu a bit confusing. Consider just a(lambda, mu)? - the introduction of \Pi^res could perhaps be deferred till after Lemma 1. - consider swapping statements 4 and 5 for lemma 1, since the proof seems to do the same. - in the proof of lemma 1, consider not having all the equations inline to improve readability. - typo: "Theorem 1 (identifiablity)" - I felt that section 4 could be more explicit about the suggested algorithm, i.e. that one simply perform the mixture modelling on the classifier outputs on the positive and unlabelled samples. - it might be worth explaining in more detail what exactly the probabilistic model summarised in Figure 1 corresponds to. That is, is it meant to represent an assumed generative view of the data, which must be satisfied for the data generation process? Or a means by which one processes the available data one has? Put another way, suppose one is able to draw from the unlabelled distribution, and then separately from the noisy positive distribution; what relevance does Fig 1 have? - is (12) related to the result about the label probability function (Proposition 1) in Ward et al. Presence-only data and the EM algorithm. Biometrics, 2009.

Confidence in this Review

2-Confident (read it all; understood it all reasonably well)


Reviewer 2

Summary

The paper presents algorithms for estimating class priors as well as posteriors from noisy positives and unlabeled data. The paper closes follows the methodology in Jain et.al., 2016; however, that work was restricted to pure positives case. The improvements over jain et.al. are illustrated in the simulations.

Qualitative Assessment

1. Recent work [1*] (see below) is not cited; whereas the authors of [1*] achieve improvement over Jain et.al., 2016. I think it will be helpful if in theory and in simulations the proposed methodology is compare to [1*]. 2. Given Jain et.al, the novelty and technicality in the proofs seems to be limited. [1*] HarishG.Ramaswamy, ClaytonScott and AmbujTewari. MixtureProportionEstimationviaKernelEmbeddingofDistributions. ICML 2016.

Confidence in this Review

2-Confident (read it all; understood it all reasonably well)


Reviewer 3

Summary

The paper introduces a method to estimate class priors in a positive and unlabeled setting. The authors account for the possibility of false (known) positives, which are very common in practice but rarely addressed in existing work. To avoid the curse of dimensionality, the proposed approach first transforms data to a single dimension using a non-traditional classifier and then proceeds with density estimation. The benchmark shows that the proposed approach works well.

Qualitative Assessment

Overall, the manuscript is well written and easy to follow. The proofs are involved and somewhat heavy on notation, but the fact the authors summarize the main practical impact of each proof greatly improves readability and flow of the text. The insights of theorem 1 and 2 are very useful. I have a few comments on the current text: - The authors mention that this is the first approach they are aware of for noisy positive and unlabeled data (lines 57-58 and 312-314). However, prior work exists that explicitly addresses this variant of learning from positive and unlabeled data [1], though it is understandable that some prior work is missed. Additionally, learning from noisy positives and unlabeled data can also be considered in the broad setting of learning with label noise (i.e., noisy positives vs noisy negatives), for which a plethora of methods is available (e.g., reviewed in [2]). Hence, I must disagree with the claims that the proposed method is the first that addresses this specific task. - What happens if the non-traditional classifier is poor (e.g., overfitting, underfitting, trivial)? How exactly is the non-traditional classifier used to transform to a single dimension, i.e., is it done in a cross-validation setup? How do you optimize hyperparameters of the non-traditional classifier (specifically, which score function do you suggest using)? Overall, the practical details of using the non-traditional classifier should be explained better. - The use of regression data sets in the benchmark seems awkward and unnecessary. I suggest replacing them with other classification problems. [1] M. Claesen, F. De Smet, J. Suykens, B. De Moor, A robust ensemble approach to learn from positive and unlabeled data using SVM base models, Neurocomputing (2015) 73-84. http://dx.doi.org/10.1016/j.neucom.2014.10.081 [2] B. Frenay, M. Verleysen, Classification in the presence of label noise: a survey, IEEE Trans. Neural Netw. Learn. Syst. 25 (May (5)) (2014) 845–869. http://dx.doi.org/10.1109/TNNLS.2013.2292894, ISSN 2162-237X.

Confidence in this Review

2-Confident (read it all; understood it all reasonably well)


Reviewer 4

Summary

A Bayesian classification routine is provided for estimation of positive only labels primary suited for applications with noise. Theoretical results are provided to demonstrate important properties of their approach. Empirical studies are performed.

Qualitative Assessment

Theoretically, the analysis provided for the problem setup is complete and tailored appropriately to the problem. It all seems correct but admittedly I did not work through the proofs line-by-line due to time constraints. The univariate transformation section on the other hand is not that innovative. This type of approach is less than ideal for dealing with large data. It is unclear how much information is really lost with this type of approach and such mechanisms are difficult to justify to practitioners. The authors claim that their approach will be robust to noise and effective for high dimensional data. The robustness claim seems to be justified theoretically, and I think the authors make a strong case for this overall. The empirical results in Table 2 also help this case, but it is hard to know for certain. Although the proposed technique wins a lot, the row to row (and even column to column) comparisons are just all over the place (Laplace distribution only). Perhaps it is saturated with too much noise. On the real results, the approach wins 19 out of 36 times and is significant 12 of those times but is also significantly beaten 11 times. I like that the authors showed both times where they were dominant and where they were not (for example, the concrete data could have been removed but wasn't). Comparisons to other work are also appropriate. All and all their approach seems to be a safe bet. The weakness of the paper is that the authors provide little practical assessment on how the algorithm scales up with n. E.g., How long did the Shuttle data take. They just mention they are susceptible to the curse of dimensionality and note that others are as well. A much more detailed analysis of this issue is expected in this venue,the paper claims this property, but this analysis in the appropriate depth was missing from the paper.

Confidence in this Review

2-Confident (read it all; understood it all reasonably well)


Reviewer 5

Summary

This paper proposed to solve the problem of estimating the class prior probabilities when one only has access to positive and unlabeled examples, albeit with some label noise. Though their experimental results show that they indeed perform better than competing methods, the paper lacked a clear motivation for why this is an important problem to solve. Thus, even though I do not have any particular gripe against the paper, it is difficult for me to recommend this for acceptance. I do, however, fully acknowledge my lack of visibility on this topic and I would be perfectly happy with the authors or the other reviewers convincing me otherwise.

Qualitative Assessment

This paper tackles the issue of identifying the prior probability of an example being positive given that one has only access to positively labeled and unlabeled examples, with the further assumption that some examples have been mislabeled. The algorithm works by assuming each of the two distributions to be a mixture of two based distributions (the one for positive and the one for negative examples) with different mixture weights. It then jointly learns all the parameters in the mixture, either parametrically using GMMs or nonparametrically using AlphaMax (which I am not familiar with). Further, the paper proves sufficient and necessary conditions for identifiability of the class priors. Finally, they show on a wide range of datasets that the introduced models perform significantly better than competing methods. I am torn with this paper. On the one hand, there is nothing wrong with it. The previous work is mentioned, the theorems are clear and so are the proposed methods. The experiments are also plentiful. However, I have a few gripes which make me lean towards rejection. As said in the summary, I would be happy to be proven wrong and change my rating. First, and this is the major one, there is very little justification for the importance of the problem this paper is trying to solve. Though molecular biology and social networks are mentioned as domains where such data could occur, there is no compelling argument as to which problems are solved by a better approximation of the class priors. This is problematic as, in its current state, I fail to see how this paper could open the way to new practical results. Second, the theorems are weak. In particular, the condition for identifiability seems extremely strong and I do not see it ever occurring in practice: there needs to exist an x for which one of the densities is 0 and the other is positive. This is even more the case when dealing with empirical distributions where either this always happens (when using the empirical distributions) or almost never happens (when using smoothed versions). Thus, as it is, I fail to see how those theorems bring an understanding to the problem that is being solved. Finally, the proposed method relies on density estimation, which is notoriously difficult in high dimension, as mentioned in the abstract. In fact, all the datasets in the experiments are low-dimensional (d < 128), which is in stark contrast with molecular biology where d is typically very large. UPDATE AFTER AUTHORS' FEEDBACK AND DISCUSSION I read the paper again after reading the authors' feedback. I had indeed missed a critical component, which was the specific way to go from the original dimension to the unidimensional problem. I apologize to the authors for that and raised my score accordingly. I still fail to see specific real-world problems where this algorithm would bring value, which keeps my "potential impact" score low. As this is my core issue, I am still leaning towards rejection but will not fight for it because of my low confidence.

Confidence in this Review

1-Less confident (might not have understood significant parts)


Reviewer 6

Summary

Suppose we are given samples from two mixtures of a pair of (unknown) distributions, with different mixing coefficients: \mu = \alpha \mu_1 + (1-\alpha) \mu_0 \nu = \beta \mu_1 + (1-\beta) \mu_0 This paper presents methods of estimating \alpha and \beta. The suggested application is for PU-learning where \mu_1 is the density of positive samples, \mu_0 is the density of negative samples, \mu is the density of the unlabeled samples and \nu of the positive-labeled samples with 1-\beta probability of error. (e.g. if \beta=0.9 then 10% of the positive-labeled samples are actually mislabeled negative samples). One nonparametric method and one parametric method are presented and compared via simulations.

Qualitative Assessment

First, the good parts: this paper addresses a problem which seems to be important and it provides a solution to it. The paper is readable and the work seems solid. My biggest issue is that this work appears to be a rather minor extension of [1] (obviously, written by the same authors): * The theoretical results seem to have some serious overlap. * The main algorithmic contribution of this paper is the AlphaMax-N algorithm, which is a small variation of AlphaMax from [1]. The authors present another parametric algorithm MSGMM-T, but it is not competitive in most cases. * The simulations use the same UCI data sets, but with added noise. I must admit I did not have the time to perform a thorough comparison with [1], so I might be wrong in my assessment. Could the authors list which parts in this work are brand new and which parts are closely related or reformulations of results from [1]? Another, more minor gripe, is that several real-world motivations are discussed, such as facebook likes and learning certain protein interactions. But no such data set is used in the simulations. Rather, standard UCI data sets are used. If this problem really has many important applications, it should be possible to obtain such a data set and it would make the paper stronger. My personal opinion is that this is a decent paper, but it does not contain enough interesting new material to warrant publication in a top conference such as NIPS. Other suggestions for improvement: 1. Section 3 has a lot of notation and technical results that make it somewhat unpleasant to decipher. But after deciphering the results are quite intuitive and appealing. I think a couple of figures with mixtures of some simple distributions (e.g. discrete multinomial) could make the ideas much clearer. 2. It would be interesting to obtain some more theoretical results regarding the AlphaMax-N. e.g. How does the error in estimating the mixing proportions decrease as the sample size goes to infinity (perhaps under some strong conditions). 3. The authors should certainly make their code available for others to use. [1] "Nonparametric semi-supervised learning of class proportions" by Jain, White, Trosset and Radivojac (2016)

Confidence in this Review

1-Less confident (might not have understood significant parts)